# Discovering Anticancer Effects of Phytochemicals on MicroRNA in the Context of Data Mining

**DOI:** 10.3390/nu17243913

**Published:** 2025-12-14

**Authors:** Yumi Sakai, Kurataka Otsuka

**Affiliations:** 1Laboratory of Biomolecular Signaling, Tokyo NODAI Research Institute, Tokyo University of Agriculture, 1-1-1 Sakuragaoka, Setagaya, Tokyo 156-8502, Japan; 2R&D Division, Kewpie Corporation, Sengawa Kewport, 2-5-7 Sengawa-cho, Chofu-shi, Tokyo 182-0002, Japan; 3Division of Molecular and Cellular Medicine, Institute of Medical Science, Tokyo Medical University, 6-7-1 Nishi-shinjyuku, Shinjuku-ku, Tokyo 160-0023, Japan; 4Division of Translational Oncology, Fundamental Innovative Oncology Core, National Cancer Center Research Institute, 5-1-1 Tsukiji, Chuo-ku, Tokyo 104-0045, Japan

**Keywords:** microRNA, phytochemical, colorectal cancer, diet, fisetin, glabridin, silibinin

## Abstract

**Background**: miRNA is linked to a variety of human diseases, including cancer. The expression levels and profiles can be related to disease prevention and the promotion of good health. Understanding the beneficial changes in miRNA expression mediated by micro- and macronutrients is vital for maintaining optimal health. However, it remains unknown which phytochemicals affect miRNA expression, thereby hindering the identification of novel dietary functions. **Methods**: We searched for and investigated novel phytochemicals that would regulate miRNAs in colon cancer using artificial intelligence. We comprehensively analyzed miRNA expression in colon cancer cell lines treated with new phytochemical candidates using next-generation sequencing. **Results**: We identified three phytochemicals (fisetin, glabridin, and silibinin) that suppressed cell proliferation and were associated with changes in cancer-related miRNA expression in colon cancer cells. The miRNA expression profiles observed in response to each phytochemical shared some common features while also displaying compound-specific miRNA signatures. Exploratory pathway analyses of fisetin, glabridin, or silibinin have shown that each affects pathways involved in tumor development, including the p53 signaling pathway, apoptosis, cellular senescence, and colorectal cancer. **Conclusions**: The use of artificial intelligence to explore candidate compounds is beneficial, leading to the discovery of new phytochemicals modulating tumor-related miRNAs. Investigating the mechanisms of action of miRNAs will be essential for understanding new functions of dietary nutrients, thereby providing further insights into the development of diet-based health promotion and disease prevention strategies.

## 1. Introduction

Some lifestyle factors, such as diet, are associated with an increased risk of several diseases, including cancer. Fruits and vegetables provide vital nutrients, including vitamins, minerals, dietary fibers, and phytochemicals, which can prevent chronic diseases and reduce mortality [1,2,3,4]. Low intake of fruits, vegetables, and whole grains is associated with an increased risk of cancer [5]. Healthy dietary profiles such as the Mediterranean diet emphasize a higher intake of plant-based foods, including vegetables, fruits, and whole grains [4]. Resveratrol is a polyphenolic phytoalexin found in berries, grapes, and peanuts [6]. Because red wine is made from grapes rich in resveratrol, it is associated with “the French paradox,” which describes the link between the consumption of red wine and a relatively low incidence of coronary heart disease in the French population [7]. Resveratrol has been investigated for its effects on sirtuin genes and longevity, as well as its antitumorigenic properties, including inhibition of tumor initiation and progression, cell cycle regulation, and apoptosis [6]. Recently, phytochemicals such as resveratrol, quercetin, and curcumin have been reported to regulate tumor-suppressive and oncogenic microRNAs (miRNAs), thereby suppressing tumor development [6]. Notably, resveratrol was one of the first food-derived natural compounds reported to modulate miRNAs in cancer, and it has since become a prototype for studying diet–miRNA associations.

miRNAs, a family of small non-coding RNAs, represent a key class of epigenetic factors that play significant roles in various biological processes, including cell differentiation and morphogenesis, by modulating gene expression [8]. They are linked to various human diseases and lifestyle-related conditions. For instance, miRNA dysregulation has been observed in patients with cancer compared with healthy participants, and miRNA expression profiles can be used to classify poorly differentiated tumors [9]. miRNAs are present in various body fluids, such as serum, plasma, amniotic fluid, breast milk, cerebrospinal fluid, colostrum, peritoneal fluid, pleural fluid, saliva, tears, and urine [10]. Recent studies have shown that early detection of cancers can be achieved by analyzing serum miRNAs [11]. It has become relatively easy to monitor disease-related changes in miRNA profiles in the blood at an early stage of disease. The ability to utilize miRNA expression patterns as a specific and sensitive marker has been investigated, with miRNA profiles in body fluids capable of acting as indicators of health [10,12]. These studies suggest that maintaining miRNA expression levels and profiles in the body is related to disease prevention and the promotion of good health.

We have previously investigated the effects of dietary compounds on miRNAs, focusing on the structural formula of resveratrol, and found that resveratrol derivatives (gnetin C, *ε*-viniferin, and piceid) regulate similar tumor-suppressive miRNAs as resveratrol [13]. These results suggest that the structural formulae help speculate on the effects of phytochemicals. Understanding beneficial changes in miRNA expression mediated by food-derived natural products is vital for maintaining optimal health and discovering novel dietary functions. However, it is not efficient to identify candidate miRNA-controlling compounds or elucidate the impact of phytochemicals on miRNA expression solely by testing individual molecules one by one. We live in a world of vast quantities of data, which makes it difficult for scientists to absorb and interpret all relevant information comprehensively. It is challenging to read, reason, learn, or make inferences from extensive, heterogeneous collections of structured data, such as tables and data cells, and unstructured data, including mass spectrometry results and sonograms.

Artificial intelligence (AI) has become integral to modern biology because it offers scalable methods for denoising, integrating, and interpreting high-dimensional data from sequencing, imaging, and multiomics assays. It can augment experimental biology, accelerate discovery, and help convert complex measurements into testable mechanistic hypotheses. In this study, we utilized cognitive computing technology, specifically IBM Watson, which incorporates medical literature, patents, genomics, and chemical and pharmacological data, to identify novel relationships and generate new hypotheses for further evaluation [14]. Watson leveraged known vocabulary to deduce the meaning of new terms based on contextual clues, recognizing terms and many of their synonyms, such as chemical diagrams, smile strings, chemical identification numbers, and generic names. For instance, Watson can extract features from published literature to create semantic similarities and help identify five new RNA-binding proteins altered in amyotrophic lateral sclerosis [15]. Here, we selected new phytochemical candidates for modulating miRNAs using AI and examined their effects on cell proliferation and miRNA expression in colon cancer cells. We identified three novel phytochemicals (fisetin, glabridin, and silibinin) that inhibit cancer cell proliferation by regulating miRNAs. Our results provide insights into the applicability of AI to large datasets and descriptions and its feasibility in combination with biological perspectives for selecting candidate compounds and verifying their functions.

## 2. Materials and Methods

### 2.1. Cell Culture

HCT116 and HT29 (ECACC) cells were cultured in Roswell Park Memorial Institute (RPMI) 1640 medium supplemented with 10% heat-inactivated fetal bovine serum (FBS) and 1% antibiotic-antimycotic (Thermo Fisher Scientific, Waltham, MA, USA) at 37 °C in 5% CO_2_. Curcumin (Tokyo Chemical Industry Co., Ltd., Tokyo, Japan, Cat. No. C2302), fisetin (Selleck Chemicals, Houston, TX, USA, Cat. No. S2298), glabridin (FUJIFILM Wako Pure Chemical Corporation, Tokyo, Japan, Cat. No. 070-04841), kojic acid (Tokyo Chemical Industry Co., Ltd., Cat. No. K0010), naringin (FUJIFILM Wako Pure Chemical Corporation, Cat. No. 148-06371), silibinin (Tokyo Chemical Industry Co., Ltd., Cat. No. C2302), theaflavin (Tokyo Chemical Industry Co., Ltd., Cat. No. T3962), quercetin (Tokyo Chemical Industry Co., Ltd. Cat. No. P0042), and *trans*-resveratrol (FUJIFILM Wako Pure Chemical Corporation, Cat. No. 185-01721) were initially tested over a concentration range of 2.5–100 µM in cell proliferation assays to generate dose–response curves. Based on these results, we selected a concentration for each compound and cell line that corresponded to the relative cell viability for subsequent experiments.

### 2.2. Cell Proliferation Assay

Two thousand cells per well in HCT116 cells and 5000 cells per well in HT29 were seeded into 96-well plates and cultured for 72 h. Cell viability was measured using the CellTiter-Glo 2.0 kit (Promega, Madison, WI, USA) according to the manufacturer’s instructions. Luminescence was measured using a SpectraMax iD3 plate reader (Molecular Devices, San Jose, CA, USA). The experiments were independently repeated at least three times.

### 2.3. DNA Library Construction and Sequencing Analysis for miRNA

Thirty thousand cells per well in HCT116 cells and 50,000 cells per well in HT29 were seeded into 24-well plates and cultured with or without each compound for 48 h. These seeding densities were chosen to account for the different growth rates of the two cell lines and to obtain comparable confluence at the time of RNA collection. The concentration of each compound was set within the range of 50–80% relative cell viability observed in the cell proliferation assay. Within that range, the concentration at which the relative cell viability was consistent with compound treatment was used. HCT116 cells were treated with 50 μM resveratrol, 25 μM quercetin, 15 μM curcumin, 12 μM fisetin, 30 μM glabridin, or 50 μM silibinin. HT29 cells were treated with 100 μM resveratrol, 50 μM quercetin, 25 μM curcumin, 30 μM fisetin, 35 μM glabridin, or 70 μM silibinin. Total RNA and miRNA were extracted from the cells using QIAzol and miRNeasy Mini Kits (Qiagen, Germantown, MD, USA) according to the manufacturer’s protocol. DNA libraries were prepared from the extracted RNA using a QIAseq miRNA Library Kit (Qiagen) according to the manufacturer’s instructions. The DNA library concentrations were measured using Qubit dsDNA HS Assay Kit on a Qubit 4 Fluorometer (Thermo Fisher Scientific). The libraries were pooled at equimolar concentrations and sequenced on a NextSeq 2000 System (Illumina, San Diego, CA, USA) with single-end reads of 75 nucleotides according to the manufacturer’s instructions. Raw FASTQ files were analyzed using the miRNA Primary Quantification pipeline in the GeneGlobe Data Analysis Center (Qiagen). 3′ adapter and low-quality bases were trimmed with cut adapt, insert and unique molecular identifier (UMI) sequences were extracted, and reads with inserts <16 nt or UMIs <10 nt were discarded. A non-redundant set of insert sequences was then aligned stepwise with Bowtie to human miRBase V22 mature and hairpin miRNA sequences, other noncoding RNAs, and mRNAs/other RNAs, and a final mapping to the human reference genome (GRCh38). For each miRNA, read counts and UMI-collapsed molecule counts were exported from GeneGlobe Data Analysis Center for downstream analyses. Normalized expression levels were calculated as counts per million to account for variations in the sequencing depth across samples. To reduce the impact of extremely low-abundance miRNAs, we removed miRNAs with counts per million <5 in all samples. The resulting values were used for exploratory differential expression analyses and for generating heatmaps and other descriptive visualizations. The experiments were independently repeated three times.

### 2.4. Compound Prediction Using IBM Watson for Drug Discovery

This study leveraged the cognitive computing platform Watson for Drug Discovery (WDD) to predict novel food compounds associated with cancer-related miRNAs. WDD was accessed in November 2018. WDD is configured for life science research, drawing on a comprehensive corpus that includes over 29 million Medline abstracts, full-text journals, patents, and chemical data. It utilizes Natural Language Processing and machine learning techniques to infer novel relationships not explicitly stated in the literature. The computational pipeline consisted of two phases: comprehensive exploration and predictive analytics. A comprehensive exploration (Explore an Entity/Network) was performed to identify miRNAs associated with cancer. This identified 528 miRNAs and subsequently, 76 associated food components that formed the initial training set. The candidate set comprised a list of food compounds, antioxidants, and polyphenols. The WDD Predictive Analytics module was used to predict novel biological activity by assessing semantic similarity between entities. Given the dispersion of the initial training set, the data were refined into four optimized subsets. Resveratrol, quercetin, and curcumin were set as queries to guide the text and chemical predictive model. These subsets were selected based on characteristics such as semantic proximity to core compounds (resveratrol, quercetin, or curcumin) or high affinity to five or more cancer types. Candidate compounds were ranked using a hit score, which represents the number of times they appeared similar to the features of optimized training clusters (high cancer affinity and a close distance to the query compound). From this ranked list, we selected phytochemicals for in vitro testing by combining high hit scores with practical considerations relevant to dietary use, such as commercial availability, price, chemical stability, and documented safety.

### 2.5. Statistical Analysis

Statistical analyses were performed to assess the differences between groups. In cell proliferation assays, a Welch’s one-way analysis of variance was used to compare changes in relative cell viability in each treatment condition. Post hoc analysis was performed using the Games-Howell test for between-group comparisons. For miRNA sequencing data, we performed exploratory within-group comparisons between compound-treated and untreated cells using *t*-tests on normalized counts per million values for each miRNA. For each miRNA, nominal *p*-values and Benjamini–Hochberg false discovery rate *q*-values were calculated, and both statistics are reported in the Appendix A, Appendix A, to assess the robustness of individual candidates. In this exploratory screen, miRNAs with nominal *p* <  0.05 were treated as candidate differentially expressed miRNAs and were subsequently used for descriptive analyses, including heatmaps and pathway enrichment. Analyses were performed using IBM SPSS Statistics version 29.0.2.0, EXCEL TOKEI v.8.0, (ESUMI Co., Ltd., Tokyo, Japan) and Python version 3.9.18. All statistical tests were two-sided. * *p*  <  0.05 was considered statistically significant, whereas the small RNA-seq analyses should be interpreted as hypothesis-generating rather than confirmatory.

## 3. Results

### 3.1. Identification of Novel Phytochemicals: Candidates for Controlling Cell Proliferation in Colon Cancer

In the context of discovery, AI aids quantitative predictive analytics to infer relationships for which there may not yet be explicit evidence. Watson was developed with a specific understanding of scientific terminology that enabled it to make novel connections across millions of pages of text. It can recognize terms and many of their synonyms, such as chemicals, whether expressed as a chemical diagram, smile string, or a chemical identification number [14]. It can also extract features from published literature to create semantic similarities and identify new connections between entities of interest. This suggests that we will discover new compounds because they share attributes with other compounds identified in previous studies. To search for novel phytochemicals that regulate tumor cell proliferation via miRNA modulation, we provided Watson with the text names and structural formulae for three compounds (resveratrol, quercetin, and curcumin) as queries (Figure 1). This is because mounting evidence has shown that resveratrol, quercetin, and curcumin are representative phytochemicals that exhibit anti-cancer roles in regulating tumor-suppressive and oncogenic miRNAs [4,10]. In subsequent in vitro experiments, these three phytochemicals were included as reference compounds to benchmark the effects of the newly selected candidates on cell proliferation and miRNA expression. We obtained 177 candidates using predictive analytics (Appendix A). The list includes a non-stilbene polyphenol, cyanidin, and a stilbene derivative, piceid, which we previously analyzed, suggesting that this approach is feasible for deducing the expected function [13]. Considering the strategy of promoting health through consumption, we narrowed down the list based on availability, price, stability, and history of safe use. Six phytochemicals were selected as candidates: fisetin, theaflavin, silibinin, naringin, kojic acid, and glabridin (Figure 2).

Next, we examined the anti-cancer effects of the candidates and control compounds (resveratrol, quercetin, and curcumin) on the proliferation of two colon cancer cell lines (HCT116 and HT29) before analyzing their effects on miRNA expression, as cell proliferation is associated with the expression of tumor-suppressive and oncogenic miRNAs [16]. We selected colon cancer cell lines because dietary factors, such as fruits and vegetables, are associated with the onset of colon cancer [17,18,19,20,21]. The control compounds and three of the candidates (fisetin, silibinin, and glabridin) exerted antitumor effects on colon cancer cells. However, the effects of theaflavin, naringin, and kojic acid were less pronounced than those of the other compounds (Figure 3 and Figure 4 and Appendix A). Curcumin, fisetin, and glabridin showed stronger antitumor effects than other phytochemicals. We excluded theaflavin, naringin, and kojic acid from further analysis because of their ineffectiveness in controlling colon cancer cell proliferation.

### 3.2. Effects of the Phytochemicals on Modulating miRNA Expression

We comprehensively analyzed the miRNA profiles of cancer cell lines treated with or without each compound. The concentration of each compound was determined based on a relative cell viability of 50–80% observed in the analysis, as reported previously (Figure 3, Figure 4, Figure 5 and Figure 6) [13,22,23]. miRNA analysis revealed distinct expression patterns between the phytochemically treated and untreated cells (Figure 7 and Figure 8 and Appendix A). In HCT116 cells, using a nominal *p*-value as an exploratory threshold, resveratrol induced 61 miRNAs and suppressed 42, quercetin induced 33 miRNAs and suppressed 42, curcumin induced six miRNAs and suppressed two, fisetin induced 18 miRNAs and suppressed 10, glabridin induced 15 miRNAs and suppressed 53, and silibinin induced 15 miRNAs and suppressed 147 (Appendix A and Appendix A). In HT29 cells, resveratrol induced 32 miRNAs and suppressed three, quercetin induced 59 miRNAs and suppressed 20, curcumin induced 21 miRNAs and suppressed two, fisetin induced seven miRNAs, glabridin induced eight miRNAs and suppressed two, and silibinin induced 11miRNAs and suppressed 76 (Appendix A and Appendix A). FDR *q*-values corresponding to these nominal *p*-values are provided in Appendix A.

Silibinin, quercetin, and resveratrol affected the expression levels of a relatively large number of miRNAs compared to other compounds (Figure 9, Appendix A). While sharing common characteristics, each compound had its own miRNA signature despite variations between cell lines. For example, more than half of the miRNAs were unique in response to silibinin treatment in both cell lines, suggesting that silibinin may have a different mechanism of action than other phytochemicals.

### 3.3. Biological Significance of miRNA Signatures in Each Phytochemical Treatment: Pathway Analysis

To elucidate the biological association of the miRNA signatures identified in this study, we next conducted pathway analyses for the three new phytochemicals (fisetin, glabridin, and silibinin) that modulated miRNA expression using Ingenuity Pathway Analysis (IPA) and the Kyoto Encyclopedia of Genes and Genomes (KEGG) [24,25]. We used the candidate differentially expressed miRNAs identified in the exploratory analysis in HCT116 and HT29 cells treated with each compound, as shown in Appendix A. The Ingenuity Core Analysis of Diseases and Bio-functions identified various cell proliferation and tumor-related pathways with the activation *Z*-score that made predictions about potential regulators using information about the direction of gene regulation (Figure 10 and Figure 11). In fisetin-treated HCT116 cells, the pathway of apoptosis of tumor cell lines was activated (Figure 10A). In HCT116 cells treated with silibinin, the pathways of migration of tumor cell lines and migration of cells were activated, and the pathway of quantity of muscle cell lines was inhibited (Figure 10C). In HT29 cells treated with glabridin, the pathway of invasion of tumor cell lines was inhibited (Figure 11B). In HT29 cells treated with silibinin, the activation *Z*-score was not calculated for this data set. Additionally, KEGG pathway analysis using the miRWalk functional enrichment analysis tool showed that miRNA signatures were enriched in tumor-related pathways, such as the p53 signaling pathway, cellular senescence, and colorectal cancer (Appendix A). For instance, enrichments of pathways, including the p53 signaling pathway and apoptosis, were observed in fisetin-treated HCT116 cells, which was consistent with the IPA analysis. In HT29 cells treated with glabridin, we identified enriched pathways, including cellular senescence, and the p53 and TGFβ signaling pathways, which were associated with cancer cell invasion [26]. Enriched pathways, including the p53 signaling pathway, colorectal cancer, cellular senescence, and apoptosis, were observed in HCT116 cells treated with silibinin. IPA did not reveal any significant pathways; however, KEGG pathway analysis identified pathways related to cancer, such as the p53 signaling pathway and colorectal cancer, which are also associated with the phenotype. These results suggest that the phytochemicals identified in this study are associated with tumor development through the modulation of miRNA expression.

## 4. Discussion

Cancer is the leading cause of death in 57 countries and causes death in one in nine men and one in twelve women worldwide [27,28]. Cancer incidence and mortality rates are steadily increasing because the burden is related to multiple factors, including aging, environmental risk factors, and lifestyle habits such as diet. Colorectal cancer is the third most common cancer worldwide, with the second highest mortality rate [28]. The incidence of early onset colorectal cancer (age ≤50 years) has been increasing worldwide, particularly in high-income countries, including most European countries, North America, Australia, New Zealand, and Japan. These findings indicate that potential risk factors, such as a Western-style diet, obesity, and physical inactivity, are involved in the etiology of cancer. Previous studies have shown weaker inverse associations between dietary factors such as fruits, vitamins, whole grains, carbohydrates, and fiber, and colorectal cancer incidence [20,21]. Fruits and vegetables are rich sources of micro- and macronutrients with antitumor properties that contribute to health promotion and preventive strategies against cancer. Many reports have shown that diet-derived micronutrients affect miRNA expression levels and regulate cancer development in vitro and in vivo. However, the relationship and function between phytochemicals and miRNAs remain to be fully elucidated.

Recent advancements in AI have significantly enhanced the ability to predict new relationships from large-scale text data, enabling more complex inferential reasoning and discoveries across fields [29]. These technologies employ natural language processing, deep learning, and knowledge representation to extract semantic and contextual connections from unstructured data on various scales. IBM Watson exemplified these developments through the integration of natural language processing, information retrieval, and machine learning techniques as part of its DeepQA architecture, which allowed it to analyze natural language, identify novel associations, and provide evidence-based reasoning for both structured and unstructured datasets [30]. In healthcare, AI augments and accelerates research using data from thousands of journal articles and clinical trials, dramatically reducing analysis time and improving hypotheses, experimental designs, and data interpretation in ways that are not possible with traditional methods. In this study, we utilized DeepQA technology to effectively search for phytochemical candidates that control tumor cell proliferation via miRNA regulation, which resulted in the discovery of fisetin, glabridin, and silibinin.

Fisetin (3,7,3′,4′-tetrahydroxyflavone) is a natural flavonoid found in various fruits, vegetables, and plants such as strawberries, apples, onions, and mulberries [31]. Fisetin has attracted interest owing to its broad biological activities, including antioxidant, anti-inflammatory, and anti-aging effects, as well as its senolytic properties that help eliminate senescent cells [32,33]. Glabridin is a naturally derived prenylated isoflavan found almost exclusively in the roots of *Glycyrrhiza glabra* L. (licorice), a well-known traditional medicinal plant used globally in herbal medicine, food, and cosmetics [34]. Glabridin exhibits a wide range of biological properties, including anti-inflammatory, antioxidant, antimicrobial, bone-protective, anti-obesity, and hepatoprotective effects. Silymarin, a flavonolignan complex extracted from *Silybum marianum* (milk thistle), primarily consists of silibinin, which is considered to be the most active constituent responsible for its biological activity [35,36]. Many studies have demonstrated the anti-cancer potential of phytochemicals through the modulation of various mechanisms. Fisetin promotes apoptotic cell death through modulating the PI3K/AKT/mTOR, Wnt/β-catenin, and TRAIL-induced apoptotic pathways [33]. Glabridin inhibits cancer stem cell formation via the TGF-β/SMAD2 pathway in breast and liver cancers, and epithelial–mesenchymal transition through upregulation of E-cadherin and cancer cell migration and invasion by blocking FAK/Src activation in breast and lung cancers [37]. Silibinin exerts multifaceted anti-cancer effects by improving the tumor microenvironment [38]. It can suppress the activation of multiple oncogenic pathways, including PI3K/Akt, MAPK, NF-κB, Wnt/β-catenin, and JAK-STAT, resulting in reduced cancer cell proliferation, cell cycle arrest, and induction of apoptosis [38,39]. Several studies have reported that silibinin modulates the expression of specific miRNAs in various cancer cell lines, including miR-20b, miR-21, and miR-155 in breast cancer and miR-24a and miR-181a in gastric cancer [40,41,42]. Additionally, several studies have shown that glabridin suppresses cancer via miR-148a regulation in breast cancer and hepatocellular carcinoma [43,44,45]. However, studies investigating comprehensive changes in miRNA expression, particularly in colon cancer, remain limited. Our research provides new insights into the functional roles of fisetin, glabridin, and silibinin in colorectal cancer through their regulation of miRNAs.

Fisetin, glabridin, and silibinin shared various miRNAs with resveratrol, quercetin, and curcumin, which was reasonable because the phytochemical candidates were provided based on the query (resveratrol, quercetin, and curcumin) using structured and unstructured datasets (Figure 9, Appendix A) [14,30]. Some miRNAs showed pronounced changes in response to fisetin, glabridin, and silibinin in our exploratory analysis. For example, miR-3929, which showed one of the largest changes among the three compounds, induced apoptosis and inhibited tumor growth in an in vivo model of cervical cancer by downregulating Cripto-1 [46]. miR-3614-5p regulates cell proliferation, apoptosis, migration, and invasion as a tumor-suppressive miRNA in colorectal, lung, and prostate cancer [47,48,49]. The expression of miR-3614-5p is decreased in colorectal cancer tissues [50]. Patients with low miR-3614-5p expression had a significantly poorer prognosis and shorter overall survival than those with high miR-3614-5p expression. High expression of miR-3614-5p differentially augments the p53 pathway, which is linked to essential biological processes, including cell cycle arrest, cellular senescence, and apoptosis. miR-122-5p regulates anti-apoptotic and cell cycle proteins, such as Bcl-2, X-linked inhibitor of apoptosis proteins, and cyclin-dependent kinases (CDKs) in breast cancer [51]. miR-122-5p is also involved in cancer progression as a tumor suppressor in gastric, hepatocellular, and pancreatic cancer [52,53,54,55]. Fisetin, glabridin, and silibinin have unique miRNA signatures. For instance, fisetin controls miR-3664-3p expression in HCT116 cells, thereby decreasing drug resistance of colorectal cancer cells by targeting ABCG2, CYP3A4, MCL1, and MLH1 [56]. Additionally, it controls miR-449a in HT29 cells, which has tumor-suppressive roles in various cancers [57,58,59,60,61,62,63,64]. The functions of miRNAs regulated by fisetin were consistent with the findings of the pathway analysis, which showed their association with apoptosis, cell proliferation, and cancer invasion (Figure 10A and Figure 11A).

Among all the compounds tested in this study, miR-4443 was upregulated only upon glabridin treatment of HCT116 cells (Figure 9A, Appendix A). A previous study reported that miR-4443 decreased the proliferation and invasion of HCT116 cells by regulating NCOA1 and TRAF4 [65]. Notably, dipotassium glycyrrhizinate, another anti-inflammatory agent extracted from licorice roots, increases miR-4443 expression and suppresses migration through inhibition of the nuclear factor kappa B (NF-κB) pathway in a glioblastoma cell line [66]. These findings indicate that licorice (*Glycyrrhiza glabra* L.) root-derived phytochemicals exert potent antitumor activity by regulating miRNAs, including miR-4443. In HT29 cells, In HT29 cells, glabridin was associated with higher expression of several predicted tumor-suppressive miRNAs such as miR-148a-3p, miR-22-3p, and miR-125b-5p (Appendix A). As described in previous studies, miR-148a is regulated by glabridin, which was also observed following silibinin treatment in this study [43,44,45]. miR-22 and miR-125b-5p have been shown to act as tumor suppressors in mouse models in colon cancer [67,68]. miR-22 targets BCL9L and restrains Wnt/β-catenin signaling, thereby limiting EMT-driven invasion and tumor progression in colon cancer cells [67]. Low expression of miR-22 was observed in colorectal cancer clinical samples and highly metastatic cell lines and was associated with tumor metastasis, advanced clinical stage, and relapse [69,70]. miR-125b-5p expression is also decreased in both colorectal cancer tissues and cell lines [71]. This miRNA reinforces antitumor immunity by targeting TNFR2 and weakening regulatory T cell (Treg)-mediated immunosuppression [68,72]. These results suggest that the antitumor functions of miRNAs modulated by glabridin are associated with the negative regulation of the tumor cell invasion pathway (Figure 11B).

Silibinin treatment resulted in a marked upregulation of miR-320 family members in both HCT116 (miR-320a, -c, and -d) and HT29 (miR-320a, -b, -c, and -d) cells, which was observed exclusively after silibinin treatment (Appendix A). The miR-320 family has tumor suppressor functions in diverse types of cancers that are related to the repression of EMT, cell proliferation, and apoptosis [73,74,75]. Consistent with its downregulation in colorectal adenoma tissue, previous studies have shown the antitumor effects of the miR-320 family in colon cancer in vitro and in vivo, including anti-proliferation (miR-320a, -b, -c, and -d), anti-metastasis (miR-320a), and anti-progression (miR-320d) [74,76,77,78,79]. Notably, few reports of plant-derived natural compounds are known to regulate the miR-320 family [80]. Silibinin downregulates the expression of the key migration regulators MMP-2 and MMP-9 through the Jak2/STAT3 and MEK/ERK pathways in breast cancer [81,82]. It also suppresses migration by downregulating MMPs in mouse colon cancer CT26 cells [83]. Although pathway analysis in HCT116 cells suggested that silibinin increases the tumor migration pathway, our results, including those of previous studies, support the phenotype observed in this study, alongside the function of the miRNAs modulated by silibinin. Our data showed that silibinin regulated a greater number of miRNAs than the other phytochemicals examined, which may contribute to a broader range of biological pathways (Figure 9 and Figure 10C, Appendix A). Therefore, the overall effects should be interpreted with caution in the context of observed phenotypes.

This study has several limitations. First, the small RNA-seq analyses were performed with three biological replicates per condition, which inevitably limits the statistical power and the precision of variance estimates. Under such small-sample conditions, *t*-tests can provide only an approximate assessment of differential expression, and even more sophisticated RNA-seq frameworks such as DESeq2 or edgeR would still face challenges in accurately controlling the false discovery rate and detecting subtle changes. To increase transparency, we treated the small RNA-seq analyses as exploratory, reported both nominal *p*-values and Benjamini–Hochberg FDR *q*-values, and focused on descriptive patterns rather than strict genome-wide claims of significance. Second, because of the large number of phytochemicals and miRNAs examined, we did not perform orthogonal experimental validation for all candidate miRNAs in each treatment condition, nor did we directly test the causal contribution of individual miRNAs using gain- or loss-of-function approaches. As a result, the identified miRNA candidates and pathways should be interpreted as hypothesis-generating and will require targeted validation and mechanistic studies in independent experiments with larger sample sizes.

In addition, our strategy of using AI to expand the list of candidate phytochemicals and then screening their effects on miRNA profiles has conceptual and practical constraints. While cognitive computing is powerful for highlighting compounds that are likely to share properties with known bioactive phytochemicals, it may be biased toward structurally or functionally related molecules and thus may not readily capture completely novel chemotypes. Even within a relatively narrow category such as antioxidant components, our data indicate that some compounds exert apparent effects on miRNA expression. In contrast, others do not, underscoring the need for high-throughput experimental systems to test computational predictions empirically. However, truly comprehensive wet-lab screening across all miRNAs, doses, time points, and candidate compounds remains challenging in terms of cost and throughput. Consequently, the present work should be viewed as a first-pass screen that integrates AI-based prioritization with small-scale exploratory transcriptomic profiling.

More broadly, as with most cell-based mechanistic studies, using cancer cell lines exposed to purified phytochemicals under controlled in vitro conditions does not fully recapitulate the complexity of in vivo exposure and tissue microenvironments. Cell culture experiments typically require well-defined, often micromolar, concentrations to elicit measurable phenotypic and transcriptomic responses, whereas dietary intake results in complex pharmacokinetics, extensive first-pass metabolism, and distribution across multiple tissues, often leading to lower, more transient intracellular levels of parent compounds and their metabolites. Moreover, factors such as gut microbiota–mediated biotransformation, plasma protein binding, and interactions with other dietary components are not captured in simplified in vitro systems. These differences mean that the present findings should be interpreted primarily as evidence of potential cellular mechanisms rather than as direct quantitative predictions of dietary effects in humans. Further studies should integrate pharmacokinetic and pharmacodynamic information and employ more physiologically relevant experimental models, such as 3D cultures, co-culture systems, organoids, and in vivo models, to bridge the gap between cell-based assays and clinical or nutritional settings.

Fisetin, glabridin, and silibinin display distinct regulatory patterns for tumor-related miRNAs. From the perspective of dietary applications, fisetin is of particular interest because available preclinical data suggest higher oral bioavailability than several other phytochemicals evaluated in this study, including resveratrol, quercetin, curcumin, glabridin, and silibinin. The bioavailability of fisetin (44.1%) is considerably greater than that of resveratrol (20%), glabridin (6.63%), quercetin (3.61%), curcumin (approximately 1%), and silibinin (<1%) [84,85,86,87]. However, these estimates are derived from animal studies using purified compounds and cannot be directly extrapolated to human dietary intake or to the specific experimental conditions used here. In the absence of in vivo PK/PD data and functional validation of the miRNA mechanisms identified in this work, these observations should be interpreted as hypothesis-generating rather than as a definitive ranking of candidates. Strawberries, a widely consumed fruit, are the richest dietary source of fisetin (160 µg/g), which suggests that fisetin-rich foods may warrant further investigation as one of several potential dietary sources influencing miRNA regulation [87]. Natural dietary sources, including fresh fruits, vegetables, and medicinal herbs, contain a wide variety of beneficial phytochemicals that promote health. However, the recent elucidation of novel biological mechanisms, exemplified by miRNAs, not only diversifies our therapeutic strategies but also redefines the potential of well-established plant resources. In this study, we provided evidence that some phytochemicals suppress tumor cell proliferation by modulating cancer-related miRNAs. The direct relationship between miRNA regulation and anti-proliferative effects should be further investigated. Our findings expand the potential use of natural dietary sources for long-term health maintenance and disease prevention, thereby advancing our knowledge and revealing their hidden nutritional properties.

## 5. Conclusions

We combined an AI-assisted compound prioritization pipeline with cell-based assays and small RNA-seq to explore how food-derived phytochemicals modulate miRNA expression in colon cancer cells. We identified candidate compounds based on their semantic and chemical similarity to well-characterized phytochemicals such as resveratrol, quercetin, and curcumin, and then selected six phytochemicals for experimental testing. Among these, fisetin, glabridin, and silibinin suppressed colon cancer cell proliferation and were associated with distinct, compound-specific miRNA signatures. Exploratory pathway analyses suggested that the miRNAs modulated by these compounds are linked to tumor-related pathways, including p53 signaling, apoptosis, cellular senescence, and colorectal cancer. While small RNA-seq and in vitro dosing have limitations, our findings provide a proof-of-concept that AI-based text and chemical mining can be integrated with experimental miRNA profiling to prioritize dietary compounds with potential anticancer activity. Future work should include targeted validation of the most promising miRNA candidates, mechanistic studies in more physiologically relevant models, and integration with pharmacokinetic data to assess translational potential better.

## Figures and Tables

**Figure 1 nutrients-17-03913-f001:**
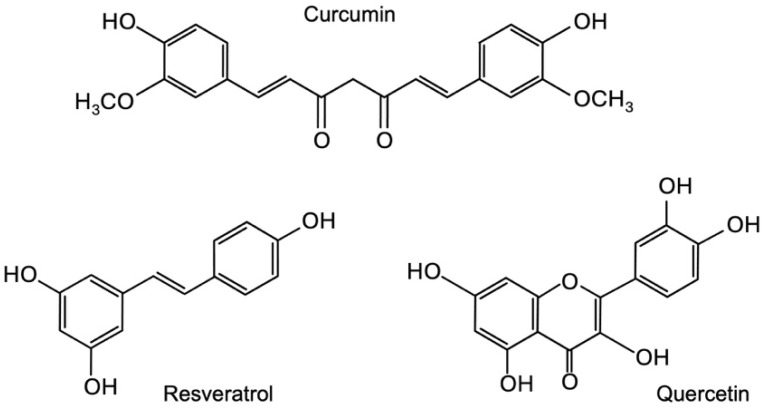
Structural formulae of curcumin, resveratrol, and quercetin.

**Figure 2 nutrients-17-03913-f002:**
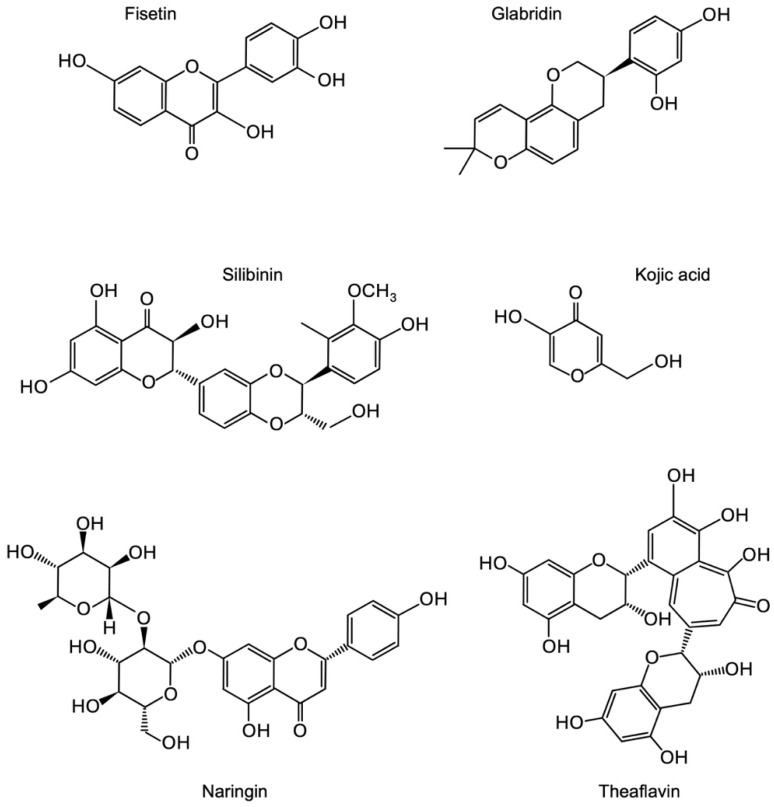
Structural formulae of phytochemical candidates.

**Figure 3 nutrients-17-03913-f003:**
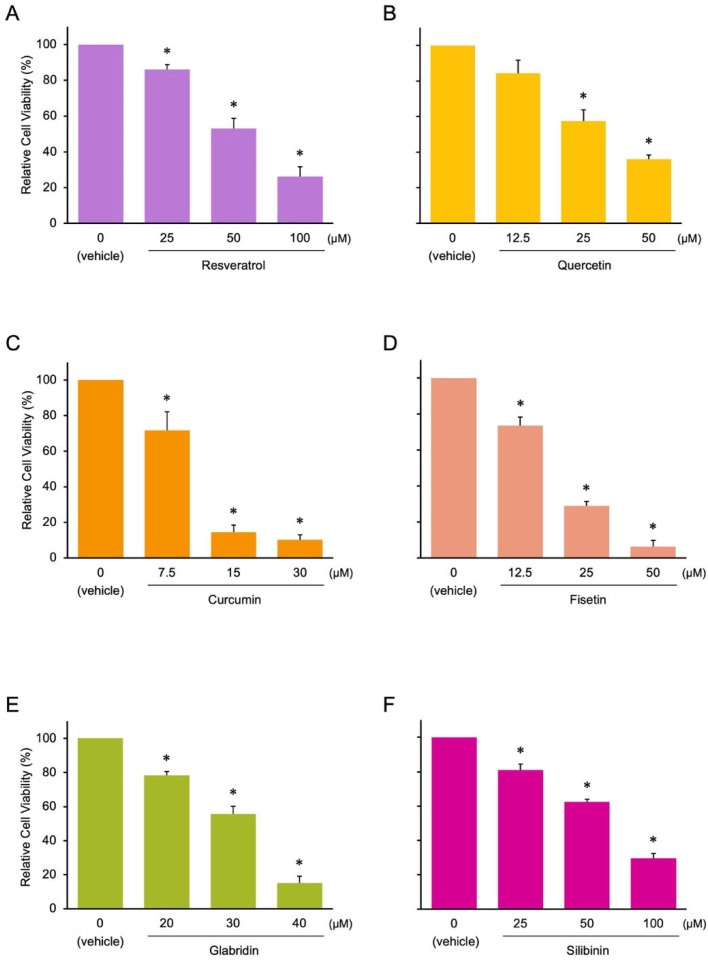
Effect of phytochemicals on the cell proliferation of HCT116 colon cancer cells. Asterisks indicate comparisons with the control group (mean ± SD, * *p* < 0.05). (**A**) Resveratrol, (**B**) Quercetin, (**C**) Curcumin, (**D**) Fisetin, (**E**) Glabridin, and (**F**) Silibinin.

**Figure 4 nutrients-17-03913-f004:**
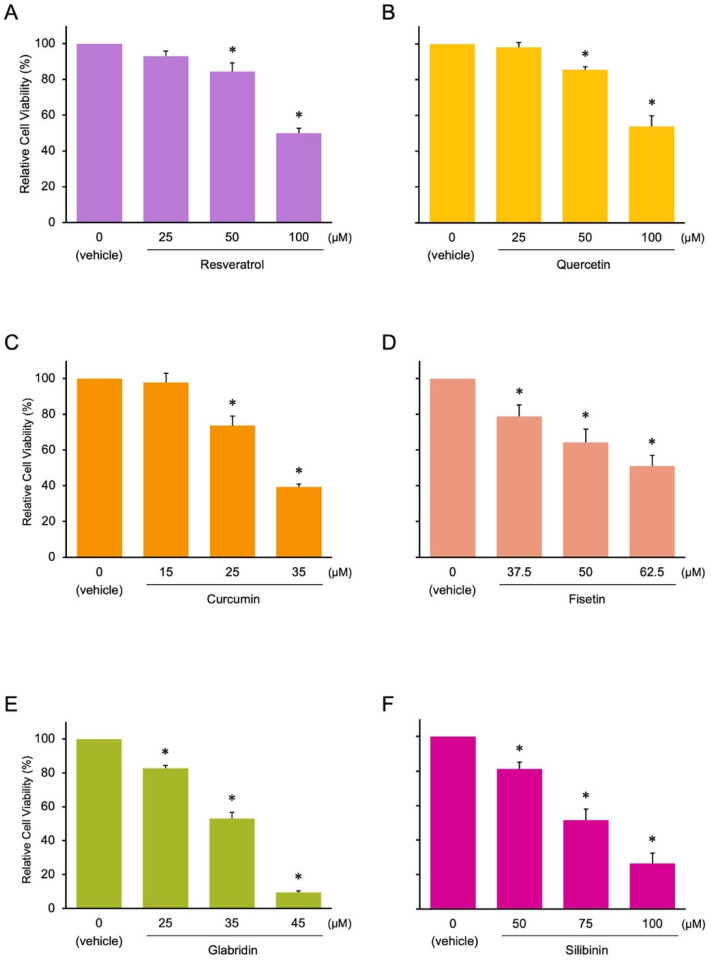
Effect of phytochemicals on the cell proliferation of HT29 colon cancer cells. Asterisks indicate comparisons with the control group (mean ± SD, * *p* < 0.05). (**A**) Resveratrol, (**B**) Quercetin, (**C**) Curcumin, (**D**) Fisetin, (**E**) Glabridin, and (**F**) Silibinin.

**Figure 5 nutrients-17-03913-f005:**
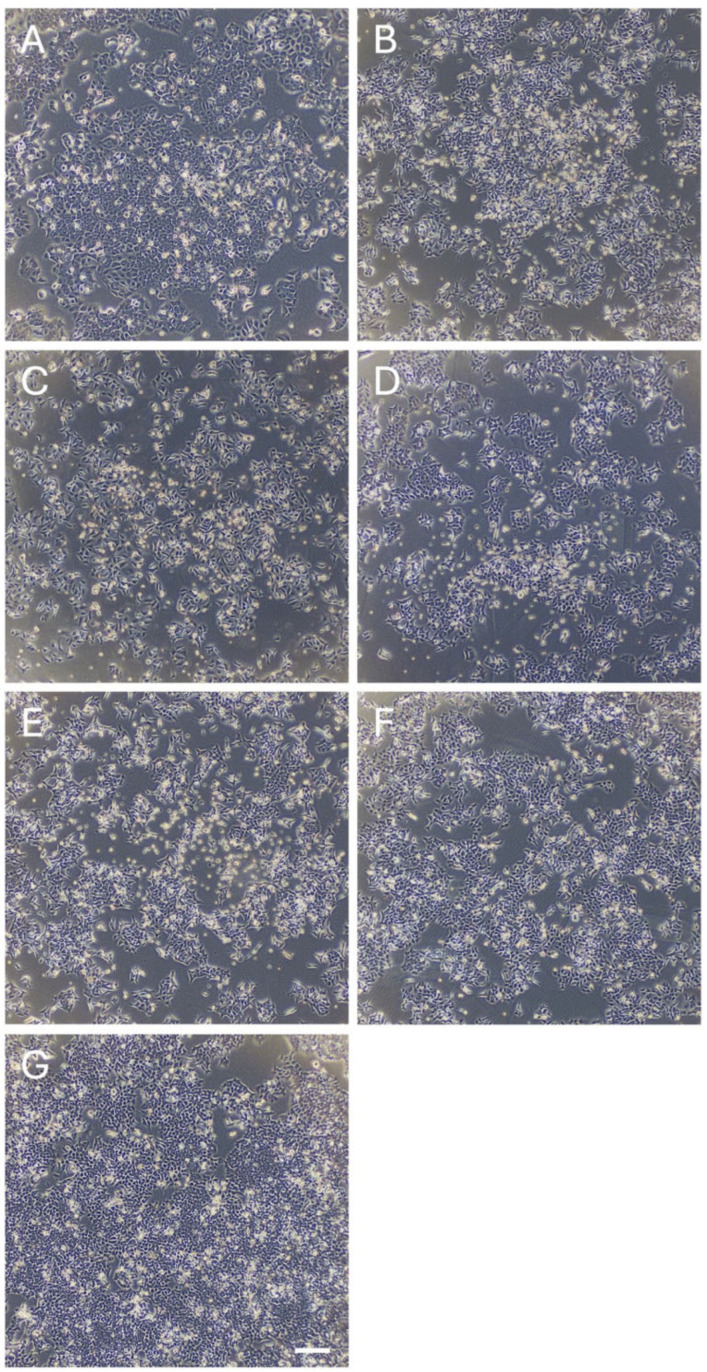
Morphology of HCT116 cells treated with or without each phytochemical. (**A**) Resveratrol, (**B**) Quercetin, (**C**) Curcumin, (**D**) Fisetin, (**E**) Glabridin, (**F**) Silibinin, and (**G**) Control (vehicle). Scale bar represents 200 µm.

**Figure 6 nutrients-17-03913-f006:**
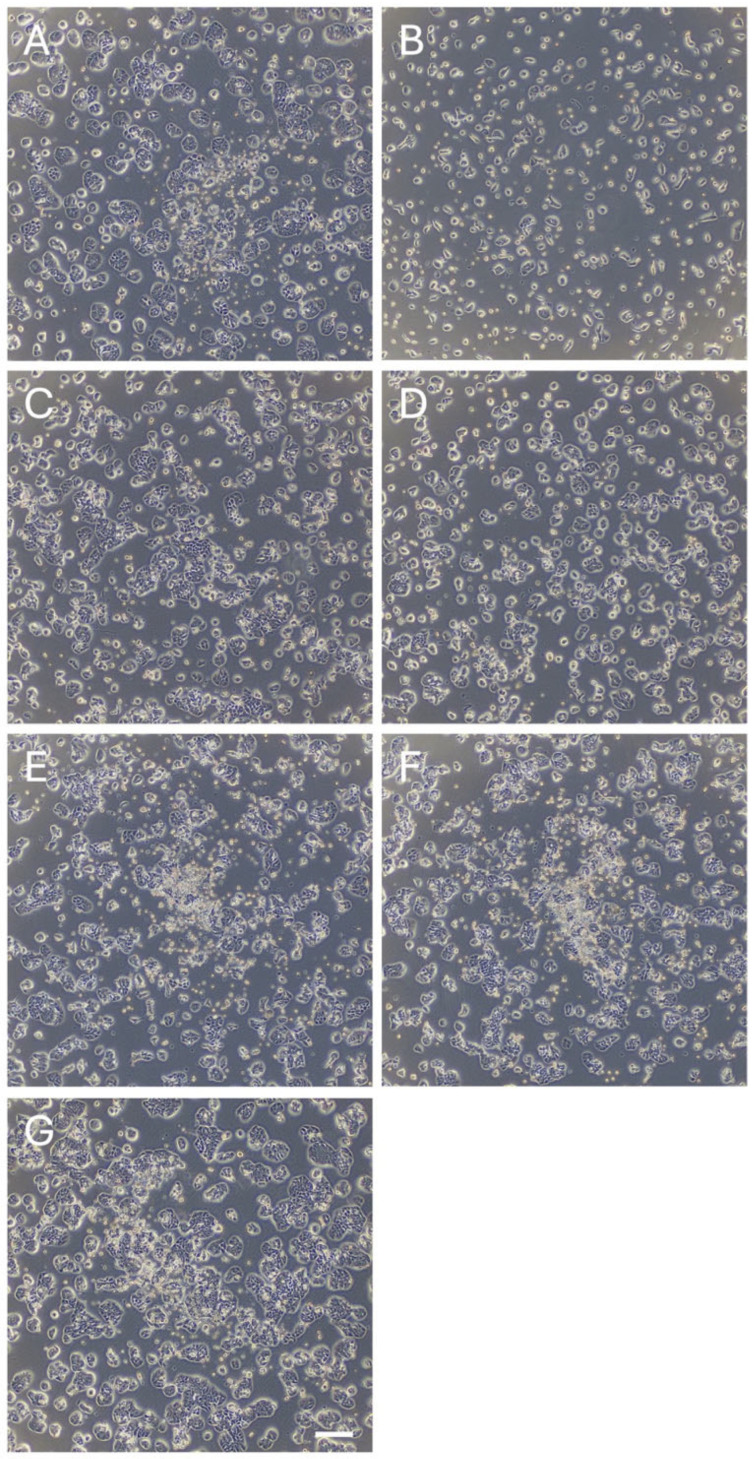
Morphology of HT29 cells treated with or without each phytochemical. (**A**) Resveratrol, (**B**) Quercetin, (**C**) Curcumin, (**D**) Fisetin, (**E**) Glabridin, (**F**) Silibinin, and (**G**) Control (vehicle). Scale bar represents 200 µm.

**Figure 7 nutrients-17-03913-f007:**
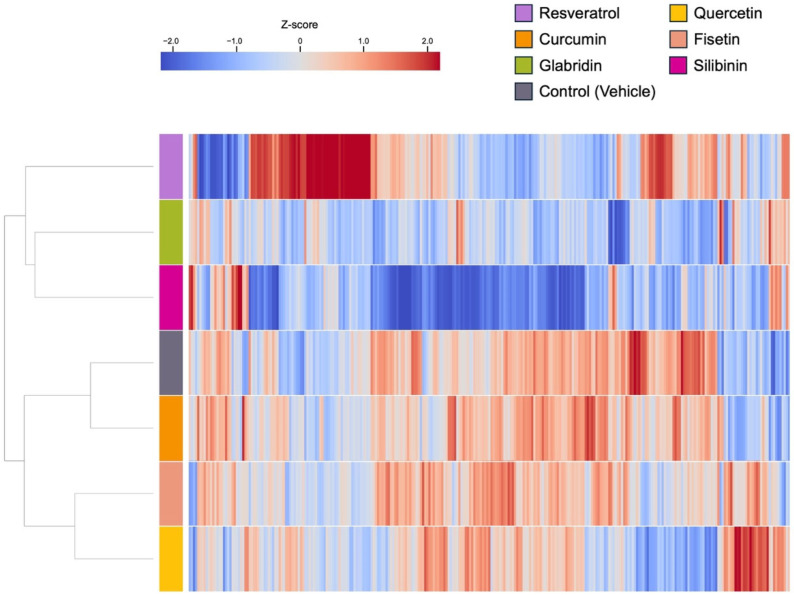
Heat map showing miRNAs that are differentially expressed in response to the treatment of each phytochemical in HCT116 cells.

**Figure 8 nutrients-17-03913-f008:**
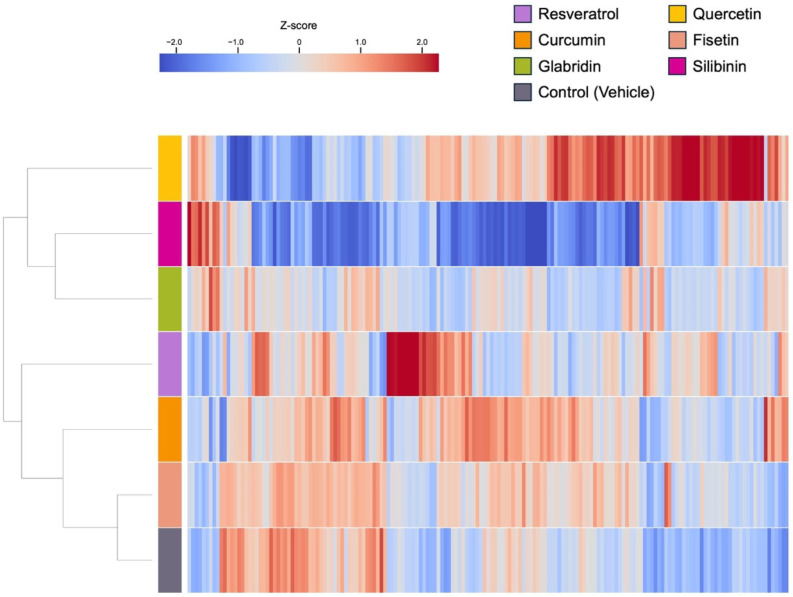
Heat map showing miRNAs that are differentially expressed in response to the treatment of each phytochemical in HT29 cells.

**Figure 9 nutrients-17-03913-f009:**
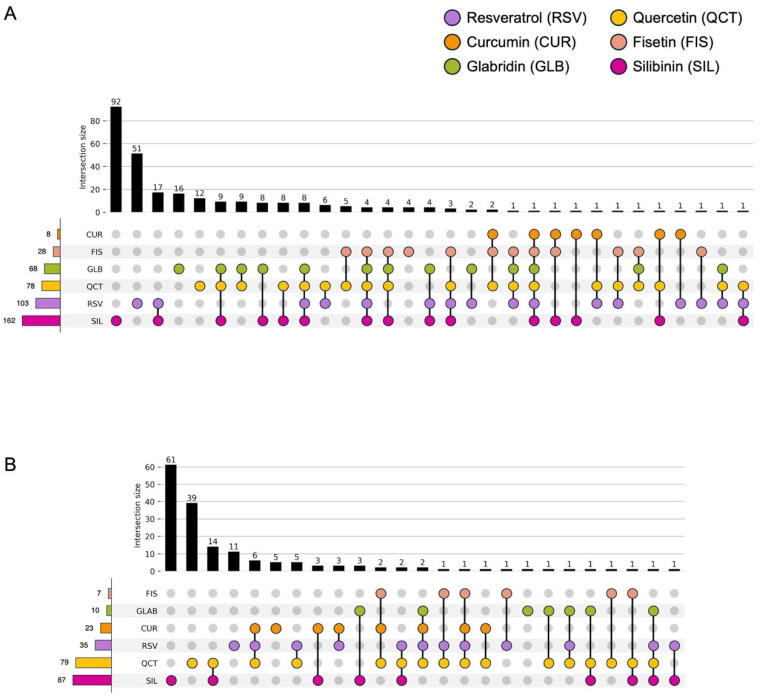
Overlapping number of miRNAs across the phytochemical treatment. (**A**) HCT116 colon cancer cell line and (**B**) HT29 colon cancer cell line.

**Figure 10 nutrients-17-03913-f010:**
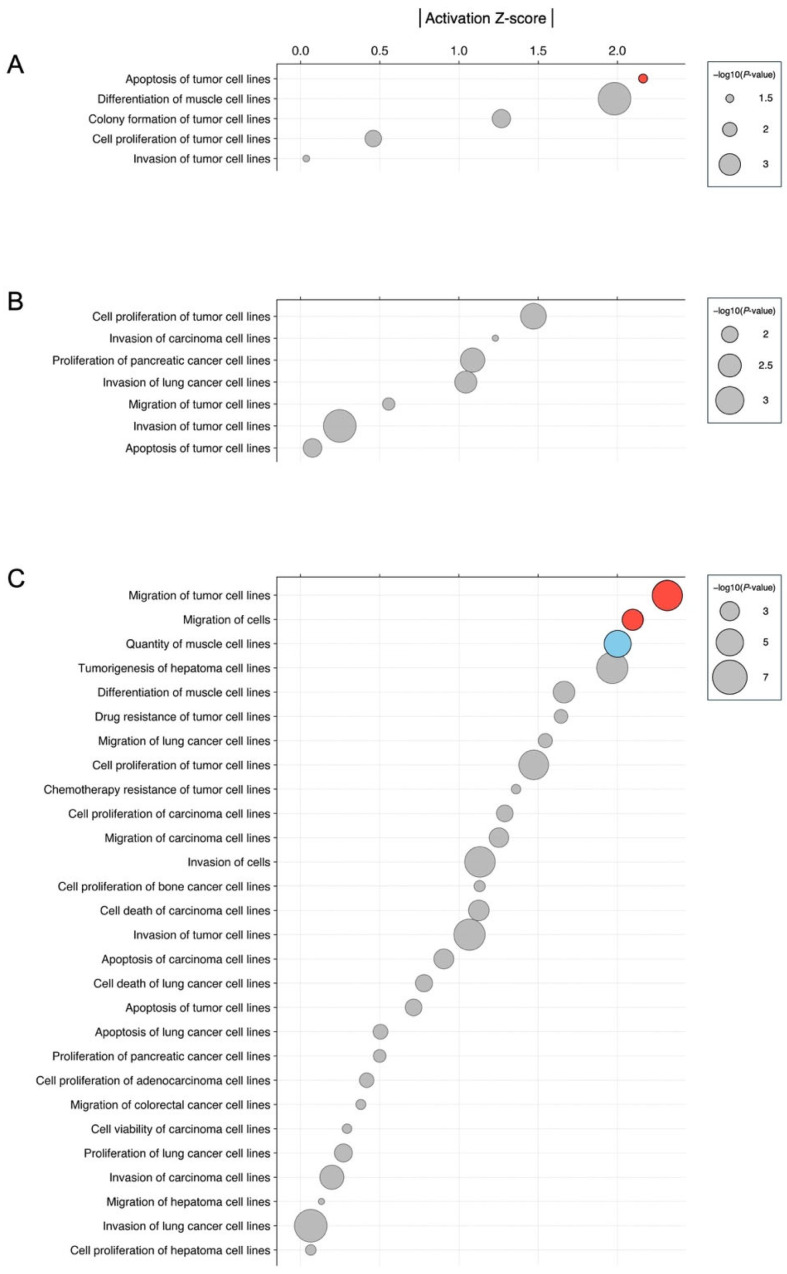
Analysis of diseases and bio-functions associated with the miRNA signature in phytochemical treatment in HCT116. (**A**) Fisetin, (**B**) Glabridin, and (**C**) Silibinin. The colors red and blue indicate activated and inhibited pathways, respectively.

**Figure 11 nutrients-17-03913-f011:**
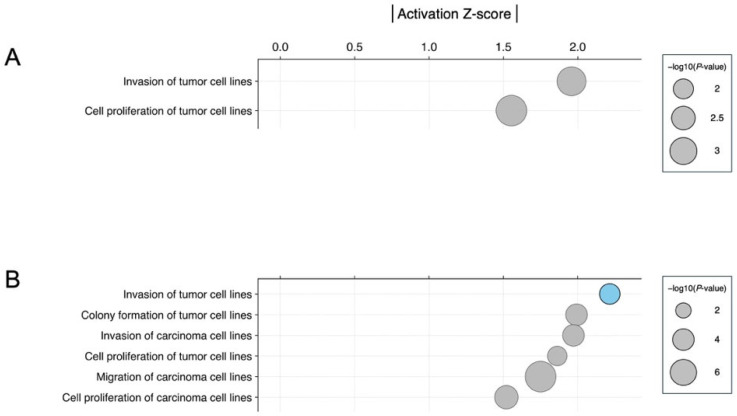
Analysis of diseases and bio-functions associated with the miRNA signature in phytochemical treatment in HT29. (**A**) Fisetin, (**B**) Glabridin, and (**C**) Silibinin. The color blue indicates an inhibited pathway.

## Data Availability

The original contributions presented in this study are included in the article/Appendix A. Further inquiries can be directed to the corresponding author.

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
