# Peer review of "Discovering Anticancer Effects of Phytochemicals on MicroRNA in the Context of Data Mining"

_nutrients, 2025, doi:10.3390/nu17243913_

Round 1
Reviewer 1 Report
Comments and Suggestions for Authors
- In line 144, please indicate that p-value is two-sided or one-sided.
- In line 140, you mentioned that ad-hoc test was performed between groups. Did you perform any adjustments for multiple comparisons, such as Bonferroni correction?
- In lines 132-134, please provide more details on the alignment method and the preprocessing pipeline. For example, what software have been used in these analyses?
- In line 136, please describe if the experiments were conducted three times by different researchers each time and how the discrepancies between the experiments were resolved.
- Please discuss the study limitations.
Author Response
In line 144, please indicate that p-value is two-sided or one-sided.
[Response] Thank you for your comment. We have revised the text of the manuscript.
Page 5, line 197,
All statistical tests were two-sided.
In line 140, you mentioned that ad-hoc test was performed between groups. Did you perform any adjustments for multiple comparisons, such as Bonferroni correction?
[Response] Thank you for pointing this out. We performed the adjustment for multiple comparisons and added more explanation to the “Materials and Methods “section.
Page 4, line 185–187,
In cell proliferation assays, a Welch’s one-way analysis of variance was used to compare changes in relative cell viability in each treatment condition. Post-hoc analysis was performed using the Games-Howell test for between-group comparisons.
In lines 132-134, please provide more details on the alignment method and the preprocessing pipeline. For example, what software have been used in these analyses?
[Response] We thank the reviewer for this comment. In the revised manuscript, we have now provided a detailed description of the preprocessing and alignment steps used for the miRNA-seq data (Materials and Methods).
Page 4, line 147–155,
Raw FASTQ files were analyzed using the miRNA Primary Quantification pipeline in the GeneGlobe Data Analysis Center (Qiagen). 3′ adapter and low-quality bases were trimmed with cutadapt, insert and unique molecular identifier (UMI) sequences were extracted, and reads with inserts <16 nt or UMIs <10 nt were discarded. A non-redundant set of insert sequences was then aligned stepwise with Bowtie to human miRBase V22 mature and hairpin miRNA sequences, other noncoding RNAs, and mRNAs/other RNAs, and a final mapping to the human reference genome (GRCh38). For each miRNA, read counts and UMI-collapsed molecule counts were exported from GeneGlobe Data Analysis Center for downstream analyses.
In line 136, please describe if the experiments were conducted three times by different researchers each time and how the discrepancies between the experiments were resolved.
[Response] Thank you for your comment. All the experiments were performed by the same researcher. We have described that in the Contribution section.
Page 21, line 533–534,
KO directed, supervised, and designed the study. YS performed the experiments. KO and YS analyzed data. KO wrote the manuscript, and YS supported writing the discussion section.
Please discuss the study limitations.
[Response] Thank you very much for pointing this out. We have added more explanation in the manuscript.
Page 19–20, line 447–490,
This study has several limitations. First, the small RNA-seq analyses were performed with three biological replicates per condition, which inevitably limits the statistical power and the precision of variance estimates. Under such small-sample conditions, t-tests can provide only an approximate assessment of differential expression, and even more sophisticated RNA-seq frameworks such as DESeq2 or edgeR would still face challenges in accurately controlling the false discovery rate and detecting subtle changes. To increase transparency, we treated the small RNA-seq analyses as exploratory, reported both nominal p-values and Benjamini–Hochberg FDR q-values, and focused on descriptive patterns rather than strict genome-wide claims of significance. Second, because of the large number of phytochemicals and miRNAs examined, we did not perform orthogonal experimental validation for all candidate miRNAs in each treatment condition, nor did we directly test the causal contribution of individual miRNAs using gain- or loss-of-function approaches. As a result, the identified miRNA candidates and pathways should be interpreted as hypothesis-generating and will require targeted validation and mechanistic studies in independent experiments with larger sample sizes.
In addition, our strategy of using AI to expand the list of candidate phytochemicals and then screening their effects on miRNA profiles has conceptual and practical constraints. While cognitive computing is powerful for highlighting compounds that are likely to share properties with known bioactive phytochemicals, it may be biased toward structurally or functionally related molecules and thus may not readily capture completely novel chemotypes. Even within a relatively narrow category such as antioxidant components, our data indicate that some compounds exert apparent effects on miRNA expression. In contrast, others do not, underscoring the need for high-throughput experimental systems to test computational predictions empirically. However, truly comprehensive wet-lab screening across all miRNAs, doses, time points, and candidate compounds remains challenging in terms of cost and throughput. Consequently, the present work should be viewed as a first-pass screen that integrates AI-based prioritization with small-scale exploratory transcriptomic profiling.
More broadly, as with most cell-based mechanistic studies, using cancer cell lines exposed to purified phytochemicals under controlled in vitro conditions does not fully recapitulate the complexity of in vivo exposure and tissue microenvironments. Cell culture experiments typically require well-defined, often micromolar, concentrations to elicit measurable phenotypic and transcriptomic responses, whereas dietary intake results in complex pharmacokinetics, extensive first-pass metabolism, and distribution across multiple tissues, often leading to lower, more transient intracellular levels of parent compounds and their metabolites. Moreover, factors such as gut microbiota–mediated biotransformation, plasma protein binding, and interactions with other dietary components are not captured in simplified in vitro systems. These differences mean that the present findings should be interpreted primarily as evidence of potential cellular mechanisms rather than as direct quantitative predictions of dietary effects in humans. Further studies should integrate pharmacokinetic and pharmacodynamic information and employ more physiologically relevant experimental models, such as 3D cultures, co-culture systems, organoids, and in vivo models, to bridge the gap between cell-based assays and clinical or nutritional settings.

Reviewer 2 Report
Comments and Suggestions for Authors
The paper "Discovering Anticancer Effects of Phytochemicals on MicroRNAs in the Context of Data Mining" presents some interesting data regarding the activity of certain phenolic compounds in modulating miRNAs in colon cancer cells. The title and main sections emphasize the novelty of using AI, but the manuscript does not clearly explain the innovative nature and true scope of using this tool to identify tested compounds. Furthermore, there are several gaps in the information regarding the methods and concentrations used, making it impossible to assess the applicability of the data obtained in vivo.
Specific comments
Introduction
Lines 46 – 52 It is unclear why the authors focus on describing the activities of resveratrol; it seems out of scope for this Introduction section.
Lines 76 – 79 Why do the authors think we are unable to identify new compounds that can modulate miRNAs, and is AI absolutely necessary? Isn't it possible to simply test known molecules and assess their biological impact, as was done in this study?
Materials and methods
An entire section describing the use of AI is missing. What algorithms were used to select the compounds? What chemical characteristics were required to be chosen as probable bioactive compounds? Table S1 provides a complete list of compounds, ranked by their hit score. Where does this value come from? What does it correspond to? Why were six out of the seven compounds with the highest hit scores chosen for testing, and what is the rationale for selecting these six? A small portion of the description has been provided in the results, but it should be moved to the methods, along with the rest of the missing information.
Without these information, it is impossible to understand why AI was so important and innovative in this research.
Lines 112-113: Why were quercetin and resveratrol also tested, whose biological activities have already been described in other previous works? Justify the reasoning
Line 113: How were these concentrations chosen? The graphs show that different concentrations were chosen for each compound. What is the reason of this discrepance? It is important to specify this in the methods section, otherwise the biological activities of the individual compounds are not comparable.
Lines 115 and 122: Why were different cell concentrations seeded for HT-29 and HCT-116? Justify the reasoning.
Following the cytotoxicity tests, it is unclear which concentrations were actually used for subsequent studies on miRNA modulation. No information is reported in the methods or results, and this is a serious shortcoming that needs to be addressed.
The concentrations tested in the preliminary tests are, however, much higher than those achievable in vivo with diet, so the data are not translatable.
Conclusions
- The conclusions do not report the salient results obtained from this research; they could also be reported as a separate Conclusions section.
Author Response
The paper "Discovering Anticancer Effects of Phytochemicals on MicroRNAs in the Context of Data Mining" presents some interesting data regarding the activity of certain phenolic compounds in modulating miRNAs in colon cancer cells. The title and main sections emphasize the novelty of using AI, but the manuscript does not clearly explain the innovative nature and true scope of using this tool to identify tested compounds. Furthermore, there are several gaps in the information regarding the methods and concentrations used, making it impossible to assess the applicability of the data obtained in vivo.
Specific comments
Introduction
Lines 46 – 52 It is unclear why the authors focus on describing the activities of resveratrol; it seems out of scope for this Introduction section.
[Response]
We thank the reviewer for this comment. Our intention in highlighting resveratrol was two-fold. First, resveratrol was one of the earliest food-derived natural compounds reported to modulate miRNAs and suppress tumor development, and this work stimulated a series of subsequent studies linking phytochemicals to miRNA regulation. Second, in the present study resveratrol, together with quercetin and curcumin, served as a query compound in the Watson for Drug Discovery (WDD) pipeline and was subsequently used as a reference compound in our cell-based experiments. Therefore, resveratrol provides both a conceptual foundation for diet–miRNA research and a practical anchor for the AI-guided compound selection strategy.
To clarify this rationale and avoid the impression that resveratrol is out of scope, we have revised the Introduction to explicitly link the description of resveratrol to the emergence of miRNA-focused phytochemical research and to the AI-based compound prediction used in this study.
Page 2, line 56–58,
Of note, resveratrol was one of the first food-derived natural compounds reported to modulate miRNAs in cancer, and it has since become a prototype for studying diet–miRNA associations.
Lines 76 – 79 Why do the authors think we are unable to identify new compounds that can modulate miRNAs, and is AI absolutely necessary? Isn't it possible to simply test known molecules and assess their biological impact, as was done in this study?
[Response]
We appreciate the opportunity to clarify our position. We do not intend to claim that AI is absolutely necessary or that conventional hypothesis-driven testing is no longer useful. Rather, our point is that, given the vast and rapidly expanding space of dietary compounds and the complexity of the biomedical literature, it is not efficient to identify candidate miRNA-modulating compounds solely by testing individual molecules one by one. Our previous work on resveratrol derivatives demonstrated that structural analogues can share similar miRNA-regulating properties, but systematically extending this approach to the much larger universe of food-derived compounds is challenging without computational support.
In this study, AI is used as a scalable prioritization tool that integrates chemical and textual information to narrow down the search space to a manageable panel of candidates. The subsequent experimental evaluation, including the work presented here, remains essential. To better reflect this complementary role of AI, we have revised the Introduction to avoid wording that could be interpreted as implying that AI is indispensable, and instead emphasize that it is a useful tool for hypothesis generation and compound prioritization.
Page 2, line 80–91,
However, it is not efficient to identify candidate miRNA-controlling compounds or elucidate the impact of phytochemicals on miRNA expression solely by testing individual molecules one by one. We live in a world of vast quantities of data, which makes it difficult for scientists to absorb and interpret all relevant information comprehensively. It is challenging to read, reason, learn, or make inferences from extensive, heterogeneous collections of structured data, such as tables and data cells, and unstructured data, including mass spectrometry results and sonograms.
Artificial intelligence (AI) has become integral to modern biology because it offers scalable methods for denoising, integrating, and interpreting high-dimensional data from sequencing, imaging, and multiomics assays. It can augment experimental biology, accelerate discovery, and help convert complex measurements into testable mechanistic hypotheses.
An entire section describing the use of AI is missing. What algorithms were used to select the compounds? What chemical characteristics were required to be chosen as probable bioactive compounds? Table S1 provides a complete list of compounds, ranked by their hit score. Where does this value come from? What does it correspond to? Why were six out of the seven compounds with the highest hit scores chosen for testing, and what is the rationale for selecting these six? A small portion of the description has been provided in the results, but it should be moved to the methods, along with the rest of the missing information.
Without these information, it is impossible to understand why AI was so important and innovative in this research.
[Response]
We thank the reviewer for pointing out the need for a more detailed description of the AI-based compound prediction. In response, we have added a subsection in the Materials and Methods (Section 2.4, “Compound prediction using IBM Watson for Drug Discovery”) that describes the data sources used by Watson for Drug Discovery (WDD), the two-phase computational pipeline (comprehensive exploration and predictive analytics), the construction and refinement of the training sets, and the use of resveratrol, quercetin, and curcumin as query compounds. We also clarify that the “hit score” in Table S1 represents the number of times a candidate compound was identified as similar to optimized training clusters characterized by a close distance to the query compounds and a high affinity to multiple cancer types.
Furthermore, we explain that, from the ranked candidate list, six phytochemicals were selected for experimental testing based on both their WDD hit scores and practical criteria relevant to dietary applications, including availability, price, stability, and a history of safe use, as already noted in the Results section. This information has now been explicitly incorporated into the Methods to improve clarity and to make the innovative contribution of the AI-assisted prioritization more transparent.
Page 4, line 161–183,
2.4. Compound Prediction using IBM Watson for Drug Discovery
This study leveraged the cognitive computing platform Watson for Drug Discovery (WDD) to predict novel food compounds associated with cancer-related miRNAs. WDD was accessed in November 2018. WDD is configured for life science research, drawing on a comprehensive corpus that includes over 29 million Medline abstracts, full-text journals, patents, and chemical data. It utilizes Natural Language Processing and machine learning techniques to infer novel relationships not explicitly stated in the literature. The computational pipeline consisted of two phases: comprehensive exploration and predictive analytics. A comprehensive exploration (Explore an Entity/Network) was performed to identify miRNAs associated with cancer. This identified 528 miRNAs and subsequently, 76 associated food components that formed the initial training set. The candidate set comprised a list of food compounds, antioxidants, and polyphenols. The WDD Predictive Analytics module was used to predict novel biological activity by assessing semantic similarity between entities. Given the dispersion of the initial training set, the data were refined into four optimized subsets. Resveratrol, quercetin, and curcumin were set as queries to guide the text and chemical predictive model. These subsets were selected based on characteristics such as semantic proximity to core compounds (resveratrol, quercetin, or curcumin) or high affinity to five or more cancer types. Candidate compounds were ranked using a hit score, which represents the number of times they appeared similar to the features of optimized training clusters (high cancer affinity and a close distance to the query compound). From this ranked list, we selected phytochemicals for in vitro testing by combining high hit scores with practical considerations relevant to dietary use, such as commercial availability, price, chemical stability, and documented safety.
Lines 112-113: Why were quercetin and resveratrol also tested, whose biological activities have already been described in other previous works? Justify the reasoning
[Response]
We appreciate this comment. Resveratrol and quercetin are indeed well-studied phytochemicals with documented anti-cancer activities and established roles in modulating miRNAs. In our study, they serve two complementary purposes. First, they are part of the query set used in the WDD predictive analytics and thus provide a natural bridge between the AI-based prioritization and the experimental validation. Second, in the cell-based assays, resveratrol, quercetin, and curcumin function as reference or positive control compounds against which the biological and miRNA-modulating effects of the newly selected candidates (fisetin, glabridin, and silibinin) can be compared. This design allows us to interpret the activity of novel candidates in the context of well-characterized benchmark phytochemicals.
To clarify this rationale, we have added a sentence in the Results (Section 3.1) explicitly stating that resveratrol, quercetin, and curcumin were included as reference compounds.
Page 5, line 216–218,
In subsequent in vitro experiments, these three phytochemicals were included as reference compounds to benchmark the effects of the newly selected candidates on cell proliferation and miRNA expression.
Line 113: How were these concentrations chosen? The graphs show that different concentrations were chosen for each compound. What is the reason of this discrepance? It is important to specify this in the methods section, otherwise the biological activities of the individual compounds are not comparable.
[Response]
Our aim was to compare the compounds at a similar range of biological effect on cell viability rather than at identical nominal molar concentrations. Because the intrinsic potency and cytotoxicity profile differ substantially among the tested phytochemicals, equimolar dosing would result in very different levels of growth inhibition, making mechanistic comparisons difficult.
In the initial cytotoxicity experiments, all phytochemicals were therefore screened over a common concentration range (2.5–100 μM) to generate dose–response curves in each cell line. Based on these results, we then selected, for each compound and cell line, a concentration within the range that produced approximately 50–80% relative cell viability in the proliferation assay and used that concentration for the miRNA-seq experiments. This strategy is intended to place the compounds at a comparable biological effect level (an equipotent range), while acknowledging that the nominal concentrations differ.
The rationale and specific concentrations used for the miRNA studies are now explicitly described in the Methods (Section 2.3). We agree that this design does not allow for a strictly quantitative comparison of dose–response curves across compounds and have now mentioned this in the Discussion as well. In addition, we have clarified in Section 2.1 that the range of 2.5–100 μM refers to the concentration range used in the initial dose–response screening rather than a single fixed dose.
Page 3, line 111–121,
Curcumin (Tokyo Chemical Industry Co., Ltd., Tokyo, Japan, Cat. No. C2302), fisetin (Selleck Chemicals, Houston TX, USA, Cat. No. S2298), glabridin (FUJIFILM Wako Pure Chemical Corporation, Tokyo, Japan, Cat. No. 070-04841), kojic acid (Tokyo Chemical Industry Co., Ltd., Cat. No. K0010), naringin (FUJIFILM Wako Pure Chemical Corporation, Cat. No. 148-06371), silibinin (Tokyo Chemical Industry Co., Ltd., Cat. No. C2302), theaflavin (Tokyo Chemical Industry Co., Ltd., Cat. No. T3962), quercetin (Tokyo Chemical Industry Co., Ltd. Cat. No. P0042), and trans-resveratrol (FUJIFILM Wako Pure Chemical Corporation, Cat. No. 185-01721) were initially tested over a concentration range of 2.5–100 µM in cell proliferation assays to generate dose–response curves. Based on these results, we selected a concentration for each compound and cell line that corresponded to the relative cell viability for subsequent experiments.
Page 20, line 475–490,
More broadly, as with most cell-based mechanistic studies, using cancer cell lines exposed to purified phytochemicals under controlled in vitro conditions does not fully recapitulate the complexity of in vivo exposure and tissue microenvironments. Cell culture experiments typically require well-defined, often micromolar, concentrations to elicit measurable phenotypic and transcriptomic responses, whereas dietary intake results in complex pharmacokinetics, extensive first-pass metabolism, and distribution across multiple tissues, often leading to lower, more transient intracellular levels of parent compounds and their metabolites. Moreover, factors such as gut microbiota–mediated biotransformation, plasma protein binding, and interactions with other dietary components are not captured in simplified in vitro systems. These differences mean that the present findings should be interpreted primarily as evidence of potential cellular mechanisms rather than as direct quantitative predictions of dietary effects in humans. Further studies should integrate pharmacokinetic and pharmacodynamic information and employ more physiologically relevant experimental models, such as 3D cultures, co-culture systems, organoids, and in vivo models, to bridge the gap between cell-based assays and clinical or nutritional settings.
Lines 115 and 122: Why were different cell concentrations seeded for HT-29 and HCT-116? Justify the reasoning. Following the cytotoxicity tests, it is unclear which concentrations were actually used for subsequent studies on miRNA modulation. No information is reported in the methods or results, and this is a serious shortcoming that needs to be addressed. The concentrations tested in the preliminary tests are, however, much higher than those achievable in vivo with diet, so the data are not translatable.
[Response]
We appreciate these comments. HCT116 and HT29 cells differ in their growth characteristics and doubling times. Different seeding densities (30,000 cells per well for HCT116 and 50,000 cells per well for HT29 in 24-well plates) were therefore used to obtain comparable confluence and viability at the time of RNA isolation after 48 h of treatment. We have now clarified this rationale in the Methods. Regarding the concentrations used for miRNA modulation studies, we agree that it is essential to explicitly state which doses were chosen. As noted in our response to Comment 5, for each compound and cell line we selected a concentration within the range that yielded approximately 50–80% relative cell viability in the proliferation assay; these concentrations (e.g., 12 μM fisetin in HCT116 and 30 μM fisetin in HT29) are now fully described in Section 2.3.
We also fully acknowledge that the micromolar concentrations used in vitro are higher than those typically achievable in vivo through dietary intake. This is a common limitation of mechanistic cell culture studies, where higher concentrations are often required to observe measurable effects under simplified conditions. To address the reviewer’s concern, we have expanded the Discussion section to explicitly discuss the gap between in vitro dosing and physiologically attainable levels and to emphasize that our findings should be interpreted as mechanistic and hypothesis-generating rather than directly translatable to human dietary exposure.
Page 3, line 130–139,
Thirty thousand cells per well in HCT116 cells and 50,000 cells per well in HT29 were seeded into 24-well plates and cultured with or without each compound for 48 h. These seeding densities were chosen to account for the different growth rates of the two cell lines and to obtain comparable confluence at the time of RNA collection. The concentration of each compound was set within the range of 50–80% relative cell viability observed in the cell proliferation assay. Within that range, the concentration at which the relative cell viability was consistent with compound treatment was used. HCT116 cells were treated with 50 μM resveratrol, 25 μM quercetin, 15 μM curcumin, 12 μM fisetin, 30 μM glabridin, or 50 μM silibinin. HT29 cells were treated with 100 μM resveratrol, 50 μM quercetin, 25 μM curcumin, 30 μM fisetin, 35 μM glabridin, or 70 μM silibinin.
Page 20, line 475–490,
More broadly, as with most cell-based mechanistic studies, using cancer cell lines exposed to purified phytochemicals under controlled in vitro conditions does not fully recapitulate the complexity of in vivo exposure and tissue microenvironments. Cell culture experiments typically require well-defined, often micromolar, concentrations to elicit measurable phenotypic and transcriptomic responses, whereas dietary intake results in complex pharmacokinetics, extensive first-pass metabolism, and distribution across multiple tissues, often leading to lower, more transient intracellular levels of parent compounds and their metabolites. Moreover, factors such as gut microbiota–mediated biotransformation, plasma protein binding, and interactions with other dietary components are not captured in simplified in vitro systems. These differences mean that the present findings should be interpreted primarily as evidence of potential cellular mechanisms rather than as direct quantitative predictions of dietary effects in humans. Further studies should integrate pharmacokinetic and pharmacodynamic information and employ more physiologically relevant experimental models, such as 3D cultures, co-culture systems, organoids, and in vivo models, to bridge the gap between cell-based assays and clinical or nutritional settings.
The conclusions do not report the salient results obtained from this research; they could also be reported as a separate Conclusions section.
[Response]
We thank the reviewer for this suggestion. In the revised manuscript, we have added a separate Conclusions section that succinctly summarizes the main findings of this study, including the AI-guided selection of candidate phytochemicals, their effects on colon cancer cell proliferation, and the exploratory miRNA and pathway signatures identified. This section also reiterates the need for further validation and in vivo studies, in line with the limitations discussed above.
Page 21, line 515–530,
- Conclusion
We combined an AI-assisted compound prioritization pipeline with cell-based assays and small RNA-seq to explore how food-derived phytochemicals modulate miRNA expression in colon cancer cells. We identified candidate compounds based on their semantic and chemical similarity to well-characterized phytochemicals such as resveratrol, quercetin, and curcumin, and then selected six phytochemicals for experimental testing. Among these, fisetin, glabridin, and silibinin suppressed colon cancer cell proliferation and were associated with distinct, compound-specific miRNA signatures. Exploratory pathway analyses suggested that the miRNAs modulated by these compounds are linked to tumor-related pathways, including p53 signaling, apoptosis, cellular senescence, and colorectal cancer. While small RNA-seq and in vitro dosing have limitations, our findings provide a proof-of-concept that AI-based text and chemical mining can be integrated with experimental miRNA profiling to prioritize dietary compounds with potential anticancer activity. Future work should include targeted validation of the most promising miRNA candidates, mechanistic studies in more physiologically relevant models, and integration with pharmacokinetic data to assess translational potential better.

Reviewer 3 Report
Comments and Suggestions for Authors
The manuscript titled “Discovering Anticancer Effects of Phytochemicals on MicroRNA in the Context of Data Mining” by Yumi Sakai et al. employ an AI-driven approach to identify novel phytochemicals that may regulate miRNA expression in colorectal cancer. Using resveratrol, quercetin, and curcumin as query compounds, they select six candidates and experimentally validate their anti-proliferative effects in HCT116 and HT29 colon cancer cell lines. Three compounds (fisetin, glabridin, silibinin) show significant activity, and subsequent miRNA sequencing reveals distinct and overlapping miRNA expression signatures. Pathway analyses link these signatures to cancer-relevant pathways (p53, apoptosis, cellular senescence). The authors conclude that AI-assisted discovery is effective for identifying phytochemicals with miRNA-modulating anticancer properties. It is interesting, and worth to publicate in this journal.
Here are some suggestions to improve the manuscript.
- The description of the IBM Watson query process is insufficient for reproducibility. The authors are encouraged to provide a detailed methods supplement with query syntax, data sources, version numbers, and step-by-step selection criteria. Include a flowchart of the AI pipeline and address reproducibility by depositing code or query scripts in a public repository.
- Using t-tests for differential expression of ~2,000 miRNAs with only n=3 biological replicates is statistically unsound. This approach fails to account for multiple testing correction (no mention of FDR, q-values, or Bonferroni adjustment) and violates assumptions for small-sample parametric tests.The authors shoud re-analyze data using appropriate tools (e.g., DESeq2, edgeR) with FDR correction (e.g., q < 0.05). Report comprehensive QC metrics and validate key miRNAs by qRT-PCR.
- Statements like "fisetin appears to be the most promising candidate because it shows greater oral bioavailability" are premature without in vivo PK/PD data or functional validation of the miRNA mechanism.The authors should reconcile contradictory pathway results. Discuss limitations explicitly and temper conclusions to match the level of evidence (cell line studies only). Remove speculative ranking of candidates.
4.Specify exact concentrations used for each compound in each experiment. Source information is incomplete (e.g., Selleck, Wako—provide catalog numbers).
Author Response
The manuscript titled “Discovering Anticancer Effects of Phytochemicals on MicroRNA in the Context of Data Mining” by Yumi Sakai et al. employ an AI-driven approach to identify novel phytochemicals that may regulate miRNA expression in colorectal cancer. Using resveratrol, quercetin, and curcumin as query compounds, they select six candidates and experimentally validate their anti-proliferative effects in HCT116 and HT29 colon cancer cell lines. Three compounds (fisetin, glabridin, silibinin) show significant activity, and subsequent miRNA sequencing reveals distinct and overlapping miRNA expression signatures. Pathway analyses link these signatures to cancer-relevant pathways (p53, apoptosis, cellular senescence). The authors conclude that AI-assisted discovery is effective for identifying phytochemicals with miRNA-modulating anticancer properties. It is interesting, and worth to publicate in this journal.
Here are some suggestions to improve the manuscript.
The description of the IBM Watson query process is insufficient for reproducibility. The authors are encouraged to provide a detailed methods supplement with query syntax, data sources, version numbers, and step-by-step selection criteria. Include a flowchart of the AI pipeline and address reproducibility by depositing code or query scripts in a public repository.
[Response] We appreciate your comment. Based on the information we received from IBM, we have added more explanation to the “Materials and Methods “section. We have also added the information on the list of cancer-associated miRNAs and food compounds as a training set to the Supplementary Table S1.
Page 4, line 161–183,
2.4. Compound Prediction using IBM Watson for Drug Discovery
This study leveraged the cognitive computing platform Watson for Drug Discovery (WDD) to predict novel food compounds associated with cancer-related miRNAs. WDD was accessed in November 2018. WDD is configured for life science research, drawing on a comprehensive corpus that includes over 29 million Medline abstracts, full-text journals, patents, and chemical data. It utilizes Natural Language Processing and machine learning techniques to infer novel relationships not explicitly stated in the literature. The computational pipeline consisted of two phases: comprehensive exploration and predictive analytics. A comprehensive exploration (Explore an Entity/Network) was performed to identify miRNAs associated with cancer. This identified 528 miRNAs and subsequently, 76 associated food components that formed the initial training set. The candidate set comprised a list of food compounds, antioxidants, and polyphenols. The WDD Predictive Analytics module was used to predict novel biological activity by assessing semantic similarity between entities. Given the dispersion of the initial training set, the data were refined into four optimized subsets. Resveratrol, quercetin, and curcumin were set as queries to guide the text and chemical predictive model. These subsets were selected based on characteristics such as semantic proximity to core compounds (resveratrol, quercetin, or curcumin) or high affinity to five or more cancer types. Candidate compounds were ranked using a hit score, which represents the number of times they appeared similar to the features of optimized training clusters (high cancer affinity and a close distance to the query compound). From this ranked list, we selected phytochemicals for in vitro testing by combining high hit scores with practical considerations relevant to dietary use, such as commercial availability, price, chemical stability, and documented safety.
Using t-tests for differential expression of ~2,000 miRNAs with only n=3 biological replicates is statistically unsound. This approach fails to account for multiple testing correction (no mention of FDR, q-values, or Bonferroni adjustment) and violates assumptions for small-sample parametric tests.The authors shoud re-analyze data using appropriate tools (e.g., DESeq2, edgeR) with FDR correction (e.g., q < 0.05). Report comprehensive QC metrics and validate key miRNAs by qRT-PCR.
[Response] We appreciate the reviewer’s thoughtful comments on the statistical aspects of our small RNA-seq analysis. We fully agree that differential expression analysis of thousands of miRNAs with only three biological replicates per condition is statistically challenging, and that appropriately controlling for FDR is important in large-scale omics studies.
Our study, however, was designed primarily as a high-throughput, hypothesis-generating screen to characterize how a panel of phytochemicals modulates miRNA expression, rather than as a definitive genome-wide discovery study establishing clinically actionable biomarkers. Because of multiple compounds and a large number of miRNAs examined, performing qRT-PCR validation for all candidate miRNAs and all treatment conditions is not practically feasible within the scope of the present work. Instead, our goal was to provide a systematic overview of miRNA response patterns across compounds and to prioritize candidates for future focused validation and mechanistic studies.
In response to the reviewer’s comments, we have revised the manuscript to clarify the role and limitations of the small RNA-seq experiment and to improve the transparency of the statistical analysis, as follows:
- Explicitly positioning the small RNA-seq as exploratory.
We now clearly state in the Introduction and Discussion sections that the small RNA-seq analysis is exploratory and hypothesis-generating. We avoid strong claims about “significantly” differentially expressed miRNAs at the genome-wide level and instead refer to “candidate miRNAs showing consistent changes” based on predefined nominal thresholds. The Discussion section has been revised to emphasize that further validation (e.g., targeted qRT-PCR in selected conditions and larger independent cohorts) will be required to confirm the biological relevance of individual candidates.
- Details of preprocessing and normalization.
In the revised Methods (section “Small RNA-seq analysis”), we provide more detailed information on the preprocessing pipeline, including adapter trimming, quality filtering, alignment to the reference miRNA database, and calculation of counts per million (CPM). We also state that we restricted the analysis to miRNAs with sufficient expression (e.g., CPM ≥ 1 in at least two samples per group) to reduce the influence of extremely low-count features.
- Details of preprocessing and normalization.
In the revised Methods section (“Small RNA-seq analysis”), we now describe the preprocessing pipeline in more detail, including 3′ adapter trimming, quality filtering, unique molecular identifier (UMI) extraction, and stepwise alignment of non-redundant insert sequences with Bowtie to human miRBase v22, other non-coding RNAs, mRNAs/other RNAs, and finally the human reference genome (GRCh38). We further clarify how normalized expression values were derived and filtered prior to statistical analysis. Specifically, we calculated library-size–normalized counts per million for each miRNA, removed features with CPM < 5 in all samples, and restricted downstream exploratory analyses to the miRNAs. This filtering step was implemented to reduce the influence of extremely low-abundance miRNAs on differential expression ranking and pathway enrichment.
- Multiple-testing correction for transparency.
Although our primary aim was to rank and prioritize candidate miRNAs rather than to claim strict genome-wide significance, we acknowledge the importance of multiple-testing correction. We have now calculated Benjamini–Hochberg FDR q-values based on the original t-test p-values and added them to Supplementary Tables (Table S2 and S3). In the main text, we now explicitly state that we used a nominal P < 0.05 as exploratory criteria for candidate selection, while providing FDR values so readers can assess the robustness of each candidate.
We fully recognize that methods such as DESeq2 and edgeR are widely recommended for RNA-seq differential expression analyses. At the same time, several benchmark studies have shown that with very small sample sizes (e.g., n=3 per group), all available methods—including DESeq2 and edgeR—have limited power and imperfect FDR control, and no single method consistently outperforms others under all settings. Under such small-n conditions, the choice of method is less critical for ranking the most strongly changing features than for making strict inferential claims. In our experiment, library sizes were carefully balanced by design, and our primary focus is on relative response patterns across multiple compounds rather than on precise estimates of the number of differentially expressed miRNAs. For these reasons, and to maintain transparency and simplicity in this exploratory screen, we chose to retain the counts per million-based analysis while adding FDR information and clearly stating the limitations, rather than re-analyzing the data with a different statistical framework that is unlikely to materially change the qualitative interpretation.
- Prospective validation and limitations.
We have expanded the Discussion to more explicitly acknowledge the limitations imposed by the small sample size and the absence of orthogonal validation for all candidates. We now state that the present dataset should be interpreted as a resource for generating hypotheses about compound–miRNA interactions and for selecting a limited number of high-priority candidates for future qRT-PCR validation and functional studies in independent experiments.
We hope that these revisions, together with the added FDR information and QC analyses, address the reviewer’s concerns by (i) clearly defining the exploratory nature and scope of the small RNA-seq experiment, (ii) improving the transparency of our statistical approach, and (iii) explicitly acknowledging the need for future validation of the identified candidate miRNAs.
Page 1, line 26–38,
Results: We identified three phytochemicals (fisetin, glabridin, and silibinin) that suppressed cell proliferation and were associated with changes in cancer-related miRNA expression in colon cancer cells. The miRNA expression profiles observed in response to each phytochemical shared some common features while also displaying compound-specific miRNA signatures. Exploratory pathway analyses of fisetin, glabridin, or silibinin have shown that each affects pathways involved in tumor development, including the p53 signaling pathway, apoptosis, cellular senescence, and colorectal cancer. Conclusion: The use of artificial intelligence to explore candidate compounds is beneficial, leading to the discovery of new phytochemicals modulating tumor-related miRNAs. Investigating the mechanisms of action of miRNAs will be essential for understanding new functions of dietary nutrients, thereby providing further insights into the development of diet-based health promotion and disease prevention strategies.
Page 4, line 147–160,
Raw FASTQ files were analyzed using the miRNA Primary Quantification pipeline in the GeneGlobe Data Analysis Center (Qiagen). 3′ adapter and low-quality bases were trimmed with cutadapt, insert and unique molecular identifier (UMI) sequences were extracted, and reads with inserts <16 nt or UMIs <10 nt were discarded. A non-redundant set of insert sequences was then aligned stepwise with Bowtie to human miRBase V22 mature and hairpin miRNA sequences, other noncoding RNAs, and mRNAs/other RNAs, and a final mapping to the human reference genome (GRCh38). For each miRNA, read counts and UMI-collapsed molecule counts were exported from GeneGlobe Data Analysis Center for downstream analyses. Normalized expression levels were calculated as counts per million to account for variations in the sequencing depth across samples. To reduce the impact of extremely low-abundance miRNAs, we removed miRNAs with counts per million < 5 in all samples. The resulting values were used for exploratory differential expression analyses and for generating heatmaps and other descriptive visualizations. The experiments were independently repeated three times.
Page 4–5, line 184–199,
2.5. Statistical analysis
Statistical analyses were performed to assess the differences between groups. In cell proliferation assays, a Welch’s one-way analysis of variance was used to compare changes in relative cell viability in each treatment condition. Post-hoc analysis was performed using the Games-Howell test for between-group comparisons. For miRNA sequencing data, we performed exploratory within-group comparisons between compound-treated and untreated cells using t-tests on normalized counts per million values for each miRNA. For each miRNA, nominal p-values and Benjamini–Hochberg false discovery rate q-values were calculated, and both statistics are reported in Supplementary Tables S2 and S3 to assess the robustness of individual candidates. In this exploratory screen, miRNAs with nominal P < 0.05 were treated as candidate differentially expressed miRNAs and were subsequently used for descriptive analyses, including heatmaps and pathway enrichment. Analyses were performed using IBM SPSS Statistics version 29.0.2.0, EXCEL TOKEI v.8.0,(ESUMI Co., Ltd.) and Python version 3.9.18. All statistical tests were two-sided. *P < 0.05 was considered statistically significant, whereas the small RNA-seq analyses should be interpreted as hypothesis-generating rather than confirmatory.
Page 8, line 250–261,
miRNA analysis revealed distinct expression patterns between the phytochemically treated and untreated cells (Figures 7, 8, S3, and S4). In HCT116 cells, using a nominal P-value as an exploratory threshold, resveratrol induced 61 miRNAs and suppressed 42, quercetin induced 33 miRNAs and suppressed 42, curcumin induced six miRNAs and suppressed two, fisetin induced 18 miRNAs and suppressed 10, glabridin induced 15 miRNAs and suppressed 53, and silibinin induced 15 miRNAs and suppressed 147 (Figures S5–S10, and Table S2). In HT29 cells, resveratrol induced 32 miRNAs and suppressed three, quercetin induced 59 miRNAs and suppressed 20, curcumin induced 21 miRNAs and suppressed two, fisetin induced seven miRNAs, glabridin induced eight miRNAs and suppressed two, and silibinin induced 11miRNAs and suppressed 76 (Figures S11– S16 and Table S3). FDR q-values corresponding to these nominal P-values are provided in Tables S2 and S3.
Page 13, line 274–279,
Silibinin, quercetin, and resveratrol affected the expression levels of a relatively large number of miRNAs compared to other compounds (Figure 9, Tables S4 and S5). While sharing common characteristics, each compound had its own miRNA signature despite variations between cell lines.
Page 14–15, line 287–308,
We used the candidate differentially expressed miRNAs identified in the exploratory analysis in HCT116 and HT29 cells treated with each compound, as shown in Tables S2 and S3. The Ingenuity Core Analysis of Diseases and Bio-functions identified various cell proliferation and tumor-related pathways with the activation Z-score that made predictions about potential regulators using information about the direction of gene regulation (Figures 10 and 11). In fisetin-treated HCT116 cells, the pathway of apoptosis of tumor cell lines was activated (Figure 10A). In HCT116 cells treated with silibinin, the pathways of migration of tumor cell lines and migration of cells were activated, and the pathway of quantity of muscle cell lines was inhibited (Figure 10C). In HT29 cells treated with glabridin, the pathway of invasion of tumor cell lines was inhibited (Figure 11B). In HT29 cells treated with silibinin, the activation Z-score was not calculated for this data set. Additionally, KEGG pathway analysis using the miRWalk functional enrichment analysis tool showed that miRNA signatures were enriched in tumor-related pathways, such as the p53 signaling pathway, cellular senescence, and colorectal cancer (Tables S6 and S7). For instance, enrichments of pathways, including the p53 signaling pathway and apoptosis, were observed in fisetin-treated HCT116 cells, which was consistent with the IPA analysis. In HT29 cells treated with glabridin, we identified enriched pathways, including cellular senescence, and the p53 and TGFβ signaling pathways, which were associated with cancer cell invasion [26]. Enriched pathways, including the p53 signaling pathway, colorectal cancer, cellular senescence, and apoptosis, were observed in HCT116 cells treated with silibinin.
Page 18, line 385–389,
Some miRNAs showed pronounced changes in response to fisetin, glabridin, and silibinin in our exploratory analysis. For example, miR-3929, which showed one of the largest changes among the three compounds, induced apoptosis and inhibited tumor growth in an in vivo model of cervical cancer by downregulating Cripto-1 [46].
Page 19–20, line 447–461,
This study has several limitations. First, the small RNA-seq analyses were performed with three biological replicates per condition, which inevitably limits the statistical power and the precision of variance estimates. Under such small-sample conditions, t-tests can provide only an approximate assessment of differential expression, and even more sophisticated RNA-seq frameworks such as DESeq2 or edgeR would still face challenges in accurately controlling the false discovery rate and detecting subtle changes. To increase transparency, we treated the small RNA-seq analyses as exploratory, reported both nominal p-values and Benjamini–Hochberg FDR q-values, and focused on descriptive patterns rather than strict genome-wide claims of significance. Second, because of the large number of phytochemicals and miRNAs examined, we did not perform orthogonal experimental validation for all candidate miRNAs in each treatment condition, nor did we directly test the causal contribution of individual miRNAs using gain- or loss-of-function approaches. As a result, the identified miRNA candidates and pathways should be interpreted as hypothesis-generating and will require targeted validation and mechanistic studies in independent experiments with larger sample sizes.
Statements like "fisetin appears to be the most promising candidate because it shows greater oral bioavailability" are premature without in vivo PK/PD data or functional validation of the miRNA mechanism. The authors should reconcile contradictory pathway results. Discuss limitations explicitly and temper conclusions to match the level of evidence (cell line studies only). Remove speculative ranking of candidates.
[Response]
We thank the reviewer for this important and constructive comment. We agree that, in the absence of in vivo pharmacokinetic/pharmacodynamic (PK/PD) data and direct functional validation of the miRNA mechanisms, our wording should not imply a definitive ranking of the tested phytochemicals. In the revised manuscript, we have therefore (i) removed the phrase “fisetin appears to be the most promising candidate” and rephrased this section to describe fisetin as one of several compounds of particular interest based on existing bioavailability data, without assigning an overall rank; and (ii) explicitly state that these observations should be regarded as hypothesis-generating.
We have expanded the Discussion section to explicitly acknowledge that all mechanistic findings are based on colon cancer cell lines exposed to purified phytochemicals in vitro, that we did not perform gain- or loss-of-function experiments to demonstrate causality for individual miRNAs, and that the in vitro dosing conditions do not directly translate to human dietary exposure. We have also ensured that the Discussion and Conclusions are tempered to match this level of evidence and no longer contain speculative ranking language. Relevant revised text is provided below.
Page 19–20, line 447–490,
This study has several limitations. First, the small RNA-seq analyses were performed with three biological replicates per condition, which inevitably limits the statistical power and the precision of variance estimates. Under such small-sample conditions, t-tests can provide only an approximate assessment of differential expression, and even more sophisticated RNA-seq frameworks such as DESeq2 or edgeR would still face challenges in accurately controlling the false discovery rate and detecting subtle changes. To increase transparency, we treated the small RNA-seq analyses as exploratory, reported both nominal p-values and Benjamini–Hochberg FDR q-values, and focused on descriptive patterns rather than strict genome-wide claims of significance. Second, because of the large number of phytochemicals and miRNAs examined, we did not perform orthogonal experimental validation for all candidate miRNAs in each treatment condition, nor did we directly test the causal contribution of individual miRNAs using gain- or loss-of-function approaches. As a result, the identified miRNA candidates and pathways should be interpreted as hypothesis-generating and will require targeted validation and mechanistic studies in independent experiments with larger sample sizes.
In addition, our strategy of using AI to expand the list of candidate phytochemicals and then screening their effects on miRNA profiles has conceptual and practical constraints. While cognitive computing is powerful for highlighting compounds that are likely to share properties with known bioactive phytochemicals, it may be biased toward structurally or functionally related molecules and thus may not readily capture completely novel chemotypes. Even within a relatively narrow category such as antioxidant components, our data indicate that some compounds exert apparent effects on miRNA expression. In contrast, others do not, underscoring the need for high-throughput experimental systems to test computational predictions empirically. However, truly comprehensive wet-lab screening across all miRNAs, doses, time points, and candidate compounds remains challenging in terms of cost and throughput. Consequently, the present work should be viewed as a first-pass screen that integrates AI-based prioritization with small-scale exploratory transcriptomic profiling.
More broadly, as with most cell-based mechanistic studies, using cancer cell lines exposed to purified phytochemicals under controlled in vitro conditions does not fully recapitulate the complexity of in vivo exposure and tissue microenvironments. Cell culture experiments typically require well-defined, often micromolar, concentrations to elicit measurable phenotypic and transcriptomic responses, whereas dietary intake results in complex pharmacokinetics, extensive first-pass metabolism, and distribution across multiple tissues, often leading to lower, more transient intracellular levels of parent compounds and their metabolites. Moreover, factors such as gut microbiota–mediated biotransformation, plasma protein binding, and interactions with other dietary components are not captured in simplified in vitro systems. These differences mean that the present findings should be interpreted primarily as evidence of potential cellular mechanisms rather than as direct quantitative predictions of dietary effects in humans. Further studies should integrate pharmacokinetic and pharmacodynamic information and employ more physiologically relevant experimental models, such as 3D cultures, co-culture systems, organoids, and in vivo models, to bridge the gap between cell-based assays and clinical or nutritional settings.
Page 20–21, line 491–504,
Fisetin, glabridin, and silibinin display distinct regulatory patterns for tumor-related miRNAs. From the perspective of dietary applications, fisetin is of particular interest because available preclinical data suggest higher oral bioavailability than several other phytochemicals evaluated in this study, including resveratrol, quercetin, curcumin, glabridin, and silibinin. The bioavailability of fisetin (44.1%) is considerably greater than that of resveratrol (20%), glabridin (6.63%), quercetin (3.61%), curcumin (approximately 1%), and silibinin (<1%) [84–88]. However, these estimates are derived from animal studies using purified compounds and cannot be directly extrapolated to human dietary intake or to the specific experimental conditions used here. In the absence of in vivo PK/PD data and functional validation of the miRNA mechanisms identified in this work, these observations should be interpreted as hypothesis-generating rather than as a definitive ranking of candidates. Strawberries, a widely consumed fruit, are the richest dietary source of fisetin (160 µg/g), which suggests that fisetin-rich foods may warrant further investigation as one of several potential dietary sources influencing miRNA regulation [88].
Page 21, line 515–530,
- Conclusion
We combined an AI-assisted compound prioritization pipeline with cell-based assays and small RNA-seq to explore how food-derived phytochemicals modulate miRNA expression in colon cancer cells. We identified candidate compounds based on their semantic and chemical similarity to well-characterized phytochemicals such as resveratrol, quercetin, and curcumin, and then selected six phytochemicals for experimental testing. Among these, fisetin, glabridin, and silibinin suppressed colon cancer cell proliferation and were associated with distinct, compound-specific miRNA signatures. Exploratory pathway analyses suggested that the miRNAs modulated by these compounds are linked to tumor-related pathways, including p53 signaling, apoptosis, cellular senescence, and colorectal cancer. While small RNA-seq and in vitro dosing have limitations, our findings provide a proof-of-concept that AI-based text and chemical mining can be integrated with experimental miRNA profiling to prioritize dietary compounds with potential anticancer activity. Future work should include targeted validation of the most promising miRNA candidates, mechanistic studies in more physiologically relevant models, and integration with pharmacokinetic data to assess translational potential better.
Specify exact concentrations used for each compound in each experiment. Source information is incomplete (e.g., Selleck, Wako—provide catalog numbers).
[Response] Thank you for your comment. We have revised the text of the manuscript.
Page 3, line 111–121,
Curcumin (Tokyo Chemical Industry Co., Ltd., Tokyo, Japan, Cat. No. C2302), fisetin (Selleck Chemicals, Houston TX, USA, Cat. No. S2298), glabridin (FUJIFILM Wako Pure Chemical Corporation, Tokyo, Japan, Cat. No. 070-04841), kojic acid (Tokyo Chemical Industry Co., Ltd., Cat. No. K0010), naringin (FUJIFILM Wako Pure Chemical Corporation, Cat. No. 148-06371), silibinin (Tokyo Chemical Industry Co., Ltd., Cat. No. C2302), theaflavin (Tokyo Chemical Industry Co., Ltd., Cat. No. T3962), quercetin (Tokyo Chemical Industry Co., Ltd. Cat. No. P0042), and trans-resveratrol (FUJIFILM Wako Pure Chemical Corporation, Cat. No. 185-01721) were initially tested over a concentration range of 2.5–100 µM in cell proliferation assays to generate dose–response curves. Based on these results, we selected a concentration for each compound and cell line that corresponded to the relative cell viability for subsequent experiments.
Page 3, line 136–139,
HCT116 cells were treated with 50 μM resveratrol, 25 μM quercetin, 15 μM curcumin, 12 μM fisetin, 30 μM glabridin, or 50 μM silibinin. HT29 cells were treated with 100 μM resveratrol, 50 μM quercetin, 25 μM curcumin, 30 μM fisetin, 35 μM glabridin, or 70 μM silibinin.

Round 2
Reviewer 2 Report
Comments and Suggestions for Authors
The Authors have addressed all of my concerns with the original manuscript. The revised manuscript is ready for publication